# Epigenome-wide association study of circulating interleukin-6 connects DNA methylation to immunometabolic and inflammatory health

Lucy Sinke [1] ✉, Jenny van Dongen[2,3], Thomas Delerue[4], Rory Wilson[4], Yujing Xia[5], Marian Beekman [1], Gonneke Willemsen[2,3], Christian Gieger [4,6], Christian Herder[7,8,9], Wolfgang Koenig [6,10,11], Annette Peters [4,6,12], Eco J. C. de Geus [2,3], José M. Ordovas[13], Jordana T. Bell [5], Melanie Waldenberger[4,6], Dorret I. Boomsma[2,3,14], P. Eline Slagboom [1] & Bastiaan T. Heijmans [1] ✉

Interleukin-6 (IL-6) drives metabolic and inflammatory processes central to disease. Current knowledge implicates epigenetic mechanisms in the regulation of these pathways, including through the methylation of CpG sites. This blood-based meta-analysis of three cohorts (n = 4,361) identifies 401 IL-6–associated CpGs enriched in regulatory regions and linked to key immunometabolic genes, including *AIM2*, *MTOR*, and *IL6R*. Three complementary causal inference approaches support most sites as responding to IL-6, with *SOCS3* (Suppressor of Cytokine Signalling 3) methylation statistically mediating inflammatory bowel disease risk. Notably, one CpG connected to *NFATC2IP* (Nuclear Factor of Activated T-cells 2 Interacting Protein) plausibly influences both IL-6 production and multiple immunometabolic conditions, including body mass index and type 2 diabetes. Collectively, our results map the DNA methylation landscape surrounding circulating IL-6 levels and unveil directional effects and distinct functional relationships between epigenetics and inflammation.

Interleukin-6 (IL-6) is a multifunctional cytokine and a central player in immunity, metabolism and disease[1,2]. A low-grade chronic inflammation, characterised by elevated levels of circulating IL-6, is observed in both ageing and multiple health conditions, including type 2 diabetes (T2D), rheumatoid arthritis (RA), and inflammatory bowel disease (IBD)[3–5]. Evidence from preclinical models, genome-wide association studies (GWAS), and Mendelian randomisation directly implicates IL-6 in the pathogenesis and progression of these diseases[6,7], and inhibitors of IL-6 signalling are successfully used as therapeutics[8–10]. Despite this, the molecular mechanisms leading to and effects of elevated IL-6 levels, remain incompletely understood.

Inflammatory cascades are complex and highly context-dependent[1,11]. IL-6 activation of JAK/STAT signalling, for example, can both amplify *IL6* expression in a positive feedback loop and also restrain IL-6 activity via Suppressor of Cytokine Signalling 3 (*SOCS3*). Furthermore, IL-6 production is governed by a multi-layered regulatory network, integrating transcription factors (TFs) with post-transcriptional control to ensure rapid yet reversible induction[1,2]. Such specific and tightly regulated processes can be orchestrated in part by epigenetic mechanisms, including DNA methylation (DNAm), which modifies genomic accessibility and TF binding, leading to

altered expression of target genes[12]. Yet, while epigenome-wide association studies (EWAS) have robustly defined how DNAm associates with IL-6 related phenotypes such as body mass index (BMI), IL-6 itself has been comparatively underexplored[13–15].

Here, we perform a meta-analysis of epigenome-wide associations with circulating IL-6 levels from over 4000 participants, exploring sensitivity of effects to smoking, twelve distinct immune cell types, and CRP. We comprehensively characterise the resulting DNAm signature through functional genomics, large-scale integrative analyses, and colocalisation. Finally, we apply three complementary causal inference techniques: triangulation, mediation analysis, and two-sample Mendelian randomisation (2SMR), to directionally connect CpGs to both IL-6 and a broad range of immunometabolic traits and diseases.

## Results

### Circulating IL-6 levels are associated with DNA methylation in blood

We conducted a meta-analysis of epigenome-wide associations with circulating IL-6 levels in a combined sample of 4361 individuals from three

cohorts (Table 1). Mean age ranged from 37.6 years in the NTR to 68.8 years in KORA, and the population was predominantly female (60.9%). Analyses were performed on DNAm profiled from whole blood samples, representing an accessible, metabolic tissue widely used in clinical diagnostics and comprised of a mixture of immune cell-types that produce and respond to circulating cytokines[16,17]. Summary statistics for all 412,226 tested CpGs are provided (Supplementary Data 1) and a study design overview with key findings is presented (Fig. 1).

After adjusting for age, sex, technical covariates, and six immune cell-types predicted from DNAm data, IL-6 was associated with 531 CpGs ($p_{fdr} < 0.05$). These associations were obtained following extensive quality control and correction for minor inflation ($0.94 \leq \lambda \leq 1.25$) and bias ($|\mu| \leq 0.25$) in the test statistics. Six CpGs were excluded due to between-cohort heterogeneity ($I^2 > 90\%$), comprising three positively and three inversely associated with IL-6. Both these removed CpGs, and the top 100 IL-6 associated CpGs showed consistent directions of effect, indicating that, despite inter-cohort differences, effect directions remained comparable (Supplementary Fig. 1).

Given that data preprocessing steps, cell-type proportions, and smoking can all influence EWAS findings, we evaluated the stability of IL-6 associations using sensitivity analyses. For DNAm data, outlying values were removed in line with published guidelines and prior work[18,19]. A sensitivity analysis in LLS excluding this processing step confirmed that it did not unduly influence effect sizes ($R = 0.95$, Supplementary Fig. 2a). For circulating IL-6 levels, undetectable values were left-censored, resulting in removal of 1.19% of total samples ($n = 52$). A sensitivity analysis in the cohort with the largest number of undetectable values (LLS) showed strong correlations between effect sizes using left censoring or limit of detection capping, indicating that effects here were also similar across data preprocessing strategies ($R = 0.98$, Supplementary Fig. 2b). Furthermore, base models in all three cohorts were extended by additionally adjusting for smoking or the estimated proportions of twelve immune cell-types[20]. Effect size estimates from these adjusted models were highly correlated with those from the base model ($R_{cell} = 0.96$ and $R_{smoke} = 0.97$), indicating that cell-type proportions and smoking had minimal confounding effects for most CpGs. However, for 130 CpGs IL-6 associations were no longer significant after correcting for multiple testing ($p_{fdr} \geq 0.05$), suggesting that these signals were partly attributable to variation in cell-type proportions or smoking. These CpGs were therefore excluded from the final results (Fig. 2a, b and Supplementary Data 2). In summary, our analyses uncovered 401 IL-6 associated CpGs mapping to 384 distinct genomic loci. Effect sizes ranged from −0.0143 to 0.0065, with most associations indicating an inverse relationship ($n = 282$, 70.3%; Fig. 3a–c).

## IL-6-associated CpGs are linked to immunometabolic risk and disease

To assess the broader relevance of the IL-6-associated CpGs to other phenotypes, we cross-referenced our findings against publicly available EWAS databases. Of the 401 CpGs associated with IL-6, 329 (82.0%) had been previously connected to at least one other trait. As anticipated, the largest overlap was with CRP (241 CpGs), an acute-phase protein whose hepatic production is directly upregulated by IL-6, representing a strong enrichment ($OR = 438.9$, $p_{fdr} < 2.2 \times 10^{-308}$)[21].

To investigate the bidirectional relationship between epigenetic signatures of these two inflammatory biomarkers further, we first assessed the impact of adjusting for hsCRP on IL-6 associations. Remarkably, following meta-analysis, all 401 IL-6-associated CpGs retained their association with IL-6 ($p_{fdr} < 0.05$) and effect sizes were only minimally attenuated ($R = 0.99$, Fig. 4a and Supplementary Data 3). To explore the reciprocal direction, we examined 1649 CpGs reported as associated with CRP and conducted an hsCRP EWAS meta-analysis in the three cohorts both with and without adjustment for IL-6 (Supplementary Data 4)[22–24]. Correction for IL-6 here resulted in substantial attenuation of the hsCRP-DNAm effect sizes (Fig. 4b). Nevertheless, the correlation between effects before and after adjustment remained high ($R = 0.90$), suggesting that despite the biological similarity of IL-6 and CRP and considerable overlap between their associated loci, their methylation signatures may represent a partially distinct and independent signal.

**Table 1 | Characteristics of the three cohorts included in the IL-6 EWAS meta-analysis**

| Characteristic | KORA F4 | LLS | NTR |
|---|---|---|---|
| Sample size | 799 | 668 | 2894 |
| Age, years | 68.8 ± 4.4 | 58.8 ± 6.7 | 37.6 ± 12.7 |
| Female sex | 390 (48.8) | 346 (51.8) | 1921 (66.4) |
| Smoking, current | 69 (8.6) | 85 (12.7) | 608 (21.0) |
| Smoking, never | 381 (47.7) | 205 (30.7) | 1603 (55.4) |
| hsCRP, mg/L | 1.47 (2.25) | 2.00 (2.33) | 1.41 (2.80) |
| IL-6, pg/mL | 1.51 (1.26) | 0.63 (0.43) | 1.00 (0.90) |

Values are shown as mean ± standard deviation for age (in years), as median (IQR) for IL-6 levels (in pg/mL) and high-sensitivity C-reactive protein (hsCRP; in mg/L), and as total (% female) for sex and smoking status.

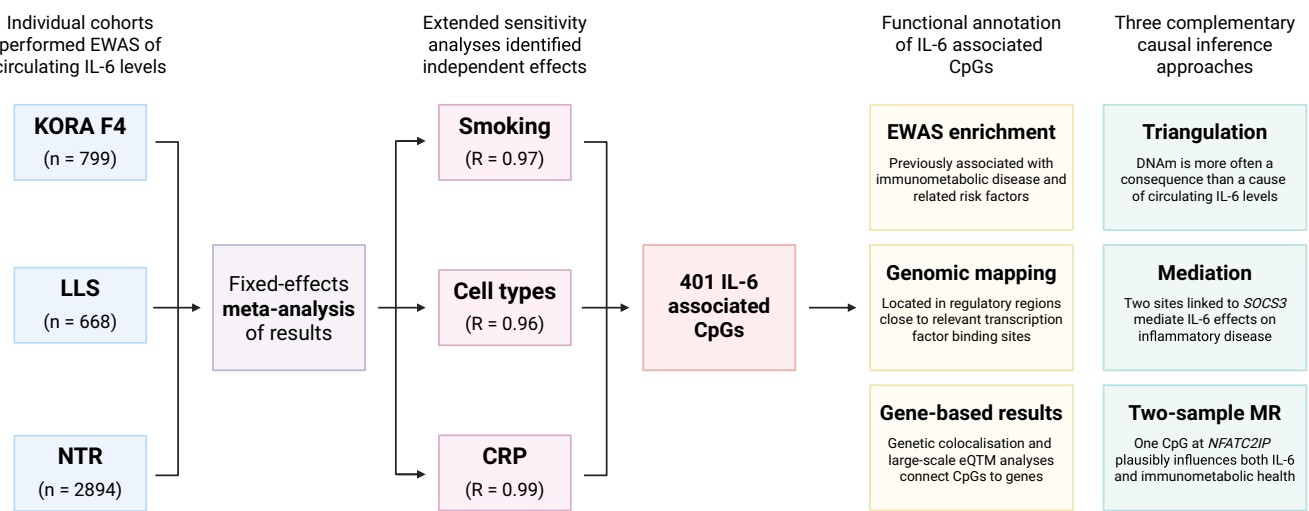

Individual cohorts performed EWAS of circulating IL-6 levels

KORA F4 (n = 799)

LLS (n = 668)

NTR (n = 2894)

Fixed-effects **meta-analysis** of results

Extended sensitivity analyses identified independent effects

**Smoking** (R = 0.97)

**Cell types** (R = 0.96)

**CRP** (R = 0.99)

**401 IL-6 associated CpGs**

Functional annotation of IL-6 associated CpGs

**EWAS enrichment**
Previously associated with immunometabolic disease and related risk factors

**Genomic mapping**
Located in regulatory regions close to relevant transcription factor binding sites

**Gene-based results**
Genetic colocalisation and large-scale eQTM analyses connect CpGs to genes

Three complementary causal inference approaches

**Triangulation**
DNAm is more often a consequence than a cause of circulating IL-6 levels

**Mediation**
Two sites linked to *SOCS3* mediate IL-6 effects on inflammatory disease

**Two-sample MR**
One CpG at *NFATC2IP* plausibly influences both IL-6 and immunometabolic health

**Fig. 1 | Flowchart summarizing the study design.** This study includes meta-analysis, extended sensitivity analyses, functional annotation, and multiple causal inference approaches. Created in BioRender.

**Fig. 2 | Sensitivity analysis of IL-6 associated CpG sites. a** Scatterplot comparing effect sizes from the base model (unadjusted) versus a model adjusted for twelve predicted blood cell-type proportions. CpG sites with insufficient evidence for an independent IL-6 effect ($p_{fdr} \geq 0.05$) after adjustment are highlighted in yellow ($n = 130$). **b** Scatterplot comparing base model effect sizes versus a model adjusted for smoking status. CpGs previously removed due to cell-type confounding are shown in grey and one CpG removed due to smoking effects is highlighted in yellow. *Data underlying* Fig. 2 *is provided in Supplementary Data 02.*

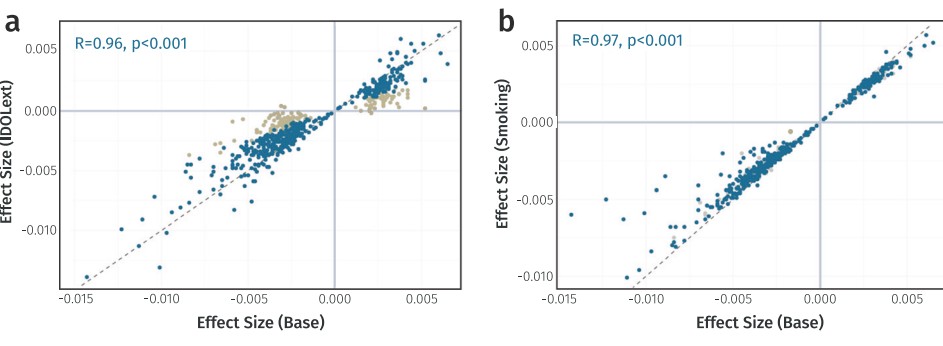

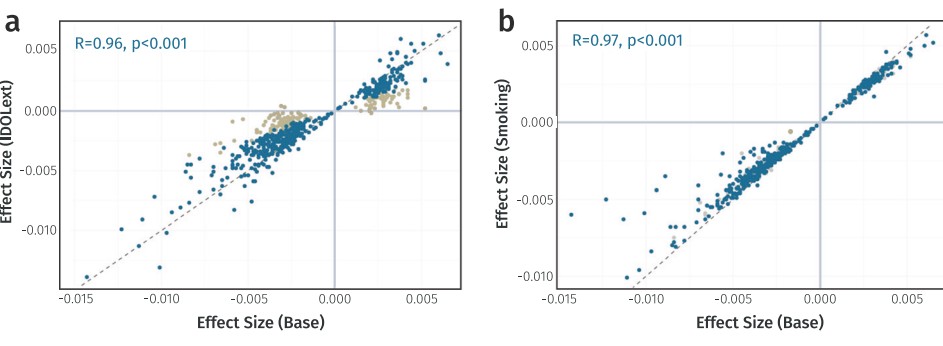

**Fig. 3 | Visualizations of DNAm associations with circulating IL-6 levels.**
**a** Directional Manhattan plot showing the genomic position of all 412,226 tested CpGs (x-axis) and their signed $\log_{10}$ p-value (y-axis). CpGs positively associated with IL-6 are displayed in the upper half, and inversely correlated CpGs in the lower half. The 401 CpGs associated with IL-6 after multiple testing correction and sensitivity analyses are highlighted in blue (odd chromosomes) and yellow (even

chromosomes), while grey dots represent all other tested CpGs. **b** Effect sizes and 95% confidence intervals (CIs) for the 401 IL-6-associated CpGs, with points coloured by mean DNAm level. **c** Volcano plot showing the effect size (x-axis) against $-\log_{10}$ p-value (y-axis) for all tested CpGs. Significant CpGs are coloured by mean DNAm level and the top hits are labelled, while all other CpGs are shown in grey. *Data underlying* Fig. 3 *is provided in Supplementary Data 01.*

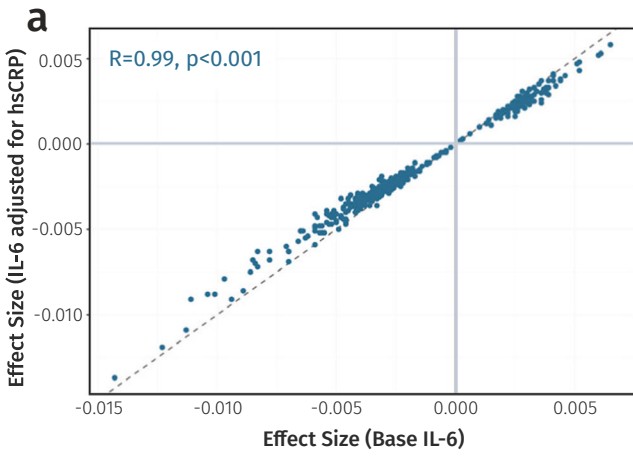
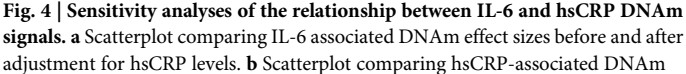
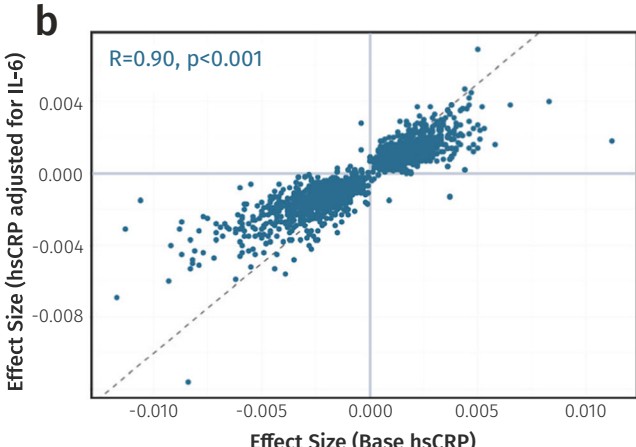

**Fig. 4 | Sensitivity analyses of the relationship between IL-6 and hsCRP DNAm signals. a** Scatterplot comparing IL-6 associated DNAm effect sizes before and after adjustment for hsCRP levels. **b** Scatterplot comparing hsCRP-associated DNAm effect sizes before and after adjustment for IL-6 levels. The Pearson correlation coefficient (R) and associated $p$-value are shown in blue for both plots. *Data underlying* Fig. 4 *is provided in Supplementary Data 01, 03, and 04.*

Overall, our analyses identified 39 traits enriched among the IL-6-associated CpGs (Fig. 5 and Supplementary Data 5). Some enrichments, such as those for education level and aggression, may have reflected interactions with environmental, social, and biological factors. For instance, education is often correlated with lifestyle factors, such as diet, which can influence systemic inflammation, while aggression may arise as a consequence of neuroinflammatory conditions[25,26]. Furthermore, although models were adjusted for age and sex, we still observed enrichment for CpGs linked to these variables. This suggested that some CpGs may have represented signatures shared between age, sex, and IL-6.

Notably, IL-6 associated CpGs were also enriched for diseases with inflammatory components, including T2D (OR = 68.0, $p_{fdr} = 7.4 \times 10^{-23}$), post-traumatic stress disorder (OR = 89.7, $p_{fdr} = 3.1 \times 10^{-56}$), and chronic kidney disease (OR = 30.0, $p_{fdr} = 1.5 \times 10^{-20}$)[27–29], as well as key immuno-metabolic risk factors, such as BMI (OR = 148.6, $p_{fdr} < 2.2 \times 10^{-308}$), lipid levels (triglyceride OR = 128.3, $p_{fdr} = 8.2 \times 10^{-33}$ and HDL cholesterol OR = 109.1, $p_{fdr} = 2.8 \times 10^{-72}$), systolic blood pressure (OR = 71.8, $p_{fdr} = 3.1 \times 10^{-48}$), fasting insulin (OR = 62.8, $p_{fdr} = 3.3 \times 10^{-87}$) and glucose (OR = 7.6, $p_{fdr} = 1.2 \times 10^{-4}$). Collectively, these enrichments pointed to a shared DNAm signature underlying IL-6 and a spectrum of inflammatory comorbidities, supporting the utility of IL-6–associated CpGs as potential biomarkers of health.

### Functional genomics highlights regulatory potential for identified CpGs

To assess regulatory context, we mapped the 401 IL-6-associated CpGs to 15 chromatin states, defined by the Roadmap Epigenomics Consortium PBMC reference epigenome (E062)[30]. These encompassed eight active and seven repressive configurations, each characterised by distinct patterns of DNAm, chromatin accessibility, and regulator binding. Our analysis revealed that IL-6-associated CpGs were enriched for four active states (Fig. 6a and Supplementary Data 6), including regions flanking active transcription (OR = 6.5, $p_{fdr} = 7.0 \times 10^{-6}$), weakly transcribed regions (OR = 1.6, $p_{fdr} = 5.8 \times 10^{-3}$), genic enhancers (OR = 3.2, $p_{fdr} = 3.2 \times 10^{-4}$), and enhancers (OR = 3.9, $p_{fdr} = 1.1 \times 10^{-18}$). Conversely, three repressive chromatin states were depleted, including polycomb repressed regions (OR = 0.48, $p_{fdr} = 5.8 \times 10^{-3}$) and bivalent transcription start sites (OR = 0.26, $p_{fdr} = 3.5 \times 10^{-2}$). These patterns indicated that IL-6-associated CpGs were preferentially located in regulatory and transcribed chromatin, rather than transcriptionally silenced regions.

Indeed, 12.7% ($n = 51$) of the IL-6 associated CpGs were annotated to enhancers in the PBMC reference, compared with 3.6% of all tested CpGs ($n = 14,767$), indicating a high probability of colocalisation with markers of open chromatin, specifically H3K4me1[31]. To evaluate cell-type specificity, we extended this analysis using reference epigenomes for 22 other immune cell types (E029-48, E050-51; Supplementary Data 7). In all tested cell types, the genomic locations of the IL-6 CpGs were enriched for enhancers ($p_{fdr} < 0.05$) demonstrating a robust regulatory potential independent of cell identity.

Since DNAm predominantly influences gene expression through TF binding modulation[32], we examined whether sequences within 50 bp of IL-6-associated CpGs were enriched for TFBS (Fig. 6b and Supplementary Data 8). Consistent with their regulatory potential, the tested regions were enriched for eleven TFBS. Notably, this included direct regulators of *IL6* expression, such as Atf4 (9 CpGs), Chop (8 CpGs), Nrf2 (3 CpGs), and subunits of the NF-κB protein complex (p50/p52: 5 CpGs; p65: 10 CpGs)[33–36]. Taken together, these analyses indicated that IL-6-associated CpGs were preferentially located within *cis*-regulatory regions active across cell-types and supported a role in immune signalling regulation.

### Integrative analyses connect IL-6-associated CpGs to expression of immunometabolic genes

To gain insight into the plausible functional consequences of this regulation, we examined correlations between DNAm at IL-6-associated CpGs and expression of genes in *cis* (±100 kb) using blood-based data from the BIOS consortium ($n = 3152$; Supplementary Data 9). Among the 1156 CpG-gene pairs evaluated, 320 (29.4%) were correlated after correction for multiple testing ($p_{fdr} < 0.05$), with most displaying an inverse relationship ($n = 255$, 75.0%). In total, 200 unique CpGs were linked to 295 unique genes, with 29 genes connected to multiple IL-6-associated CpGs. Notable examples included *SOCS3*, a negative regulator of IL-6 signalling, and the inflammasome-related genes, *IFI16* and *AIM2*[37,38].

Additionally, we leveraged large-scale quantitative trait loci (QTL) datasets to perform colocalisation analysis[39,40]. This framework evaluated whether IL-6-associated CpGs and nearby genes shared underlying genetic architecture (Supplementary Data 10). Unlike direct expression-methylation correlations, colocalisation is more robust to short-term fluctuations in expression and can reveal stable genetically driven relationships, even when transcriptional potential is not realised at time of sampling. Methylation QTL (mQTL) and expression QTL (eQTL) data were available for 914 CpG-gene pairs, of which 214 (23.4%) were flagged as colocalised, representing 130 unique CpGs and 197 unique genes.

In total, 421 genes were linked to IL-6-associated DNAm by one or both integrative analyses. Despite capturing distinct regulatory mechanisms, there was substantial overlap in CpG-gene pairs identified by both approaches ($n = 76$), indicating convergence on common biological

**Fig. 5 | Trait enrichment among IL-6-associated CpGs.** Forest plot showing natural log-transformed odds ratios (ORs) with their corresponding 95% CIs for the 20 most enriched traits. For each trait, the count and percentage of overlapping CpGs are displayed alongside the plot. *Data underlying Fig. 5 is provided in Supplementary Data 05.*

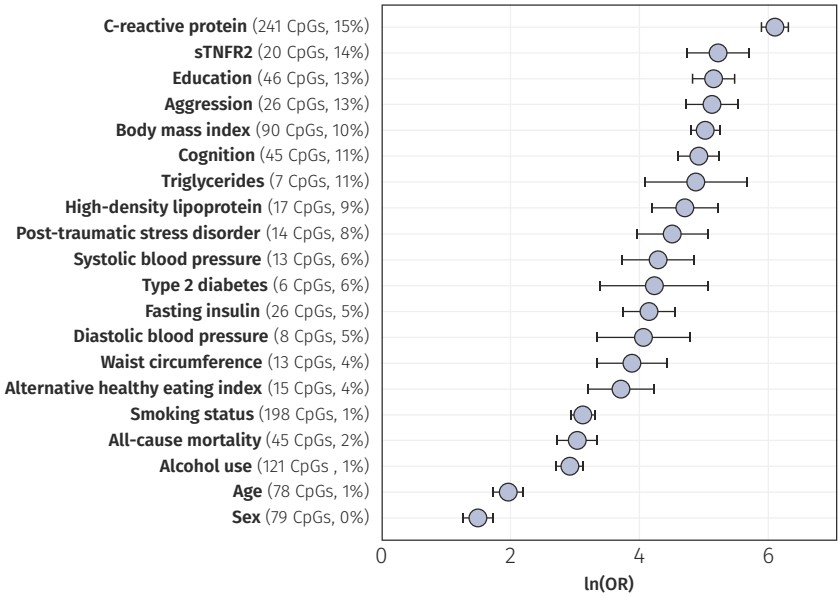

**Fig. 6 | Functional enrichment of IL-6 associated CpGs. a** Forest plot of ORs and 95% CIs with enrichment across 14 chromatin states, calculated using the Roadmap PBMC reference epigenome (E062) with all tested CpGs as the background set. States are sorted by OR, with blue denoting enrichment, yellow indicating depletion, and grey for non-significant results. One non-significant term (ZNF/Rpts) not

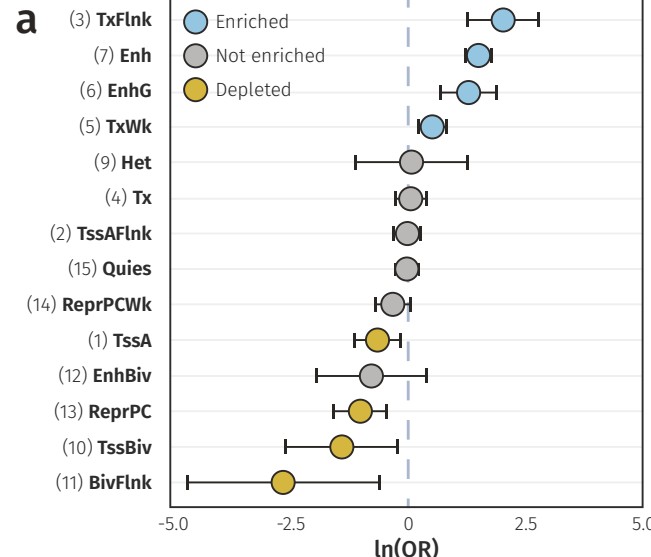
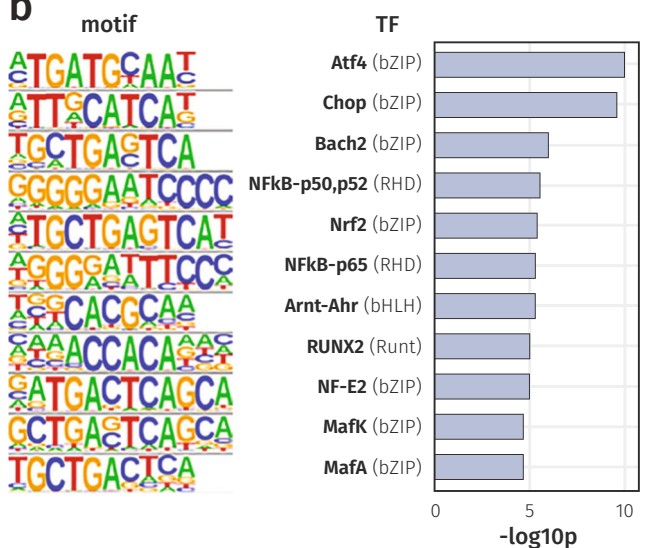

shown due to an extremely wide 95% CI. **b** Bar plot of $-\log_{10}$ *p*-values from TFBS enrichment analysis, performed using HOMER on sequences within 50 bp of IL-6 associated CpGs, tested against a GC-matched random genomic background. *Data underlying Fig. 6 is provided in Supplementary Data 06 and 08.*

patterns ($\chi^2 = 15.2$, $p = 9.9 \times 10^{-5}$). Notably, several non-nearest gene relationships were both biologically plausible and highly relevant to IL-6 biology, including the IL-6 receptor gene (*IL6R*) and polycomb group ring finger 6 (*PCGF6*), a transcriptional repressor of *IL6* in quiescent dendritic cells[41]. Together with the established regulatory capacity of IL-6-associated CpGs, these findings supported functional relevance of our results and provided a framework to confidently map IL-6-associated CpGs to plausible target genes in downstream analyses.

**Cytokine-specific pathways are linked to IL-6-associated methylation**

To systematically characterise the biological roles of the 421 genes connected to IL-6-associated DNAm, we performed an over-representation analysis. Of the 16,037 gene sets tested, 265 were enriched in our results ($p_{fdr} < 0.05$; Fig. 7a and Supplementary Data 11) with 331 of the input genes

mapping to at least one pathway (78.6%). The most strongly enriched gene set was Immune System (65 genes, $p_{fdr} = 1.0 \times 10^{-7}$), which included several key regulators of IL-6 signalling, such as *IL6R* and *SOCS3*[38,42]. Notably, although the top five terms included many large and generic sets (Immune System, Disease, and Metabolism), 25 of the enriched pathways were cytokine-specific (9.4%), including TNFα signalling via NF-κB (14 genes, $p_{fdr} = 7.2 \times 10^{-4}$), IL-2/STAT5 signalling (16 genes, $p_{fdr} = 3.9 \times 10^{-5}$), and IL-6 family signalling (4 genes, $p_{fdr} = 2.0 \times 10^{-2}$).

We next performed TF enrichment analysis using the DoRothEA database to assess whether genes linked to IL-6-associated CpGs were over-represented among known TF regulons (Fig. 7b and Supplementary Data 12). One of the top ten enriched TFs was STAT3, the canonical downstream effector of IL-6 signalling, indicating that many genes connected to IL-6-associated DNAm were direct IL-6 targets mediated through STAT3 activation[21]. Other highly enriched TFs included ETS1

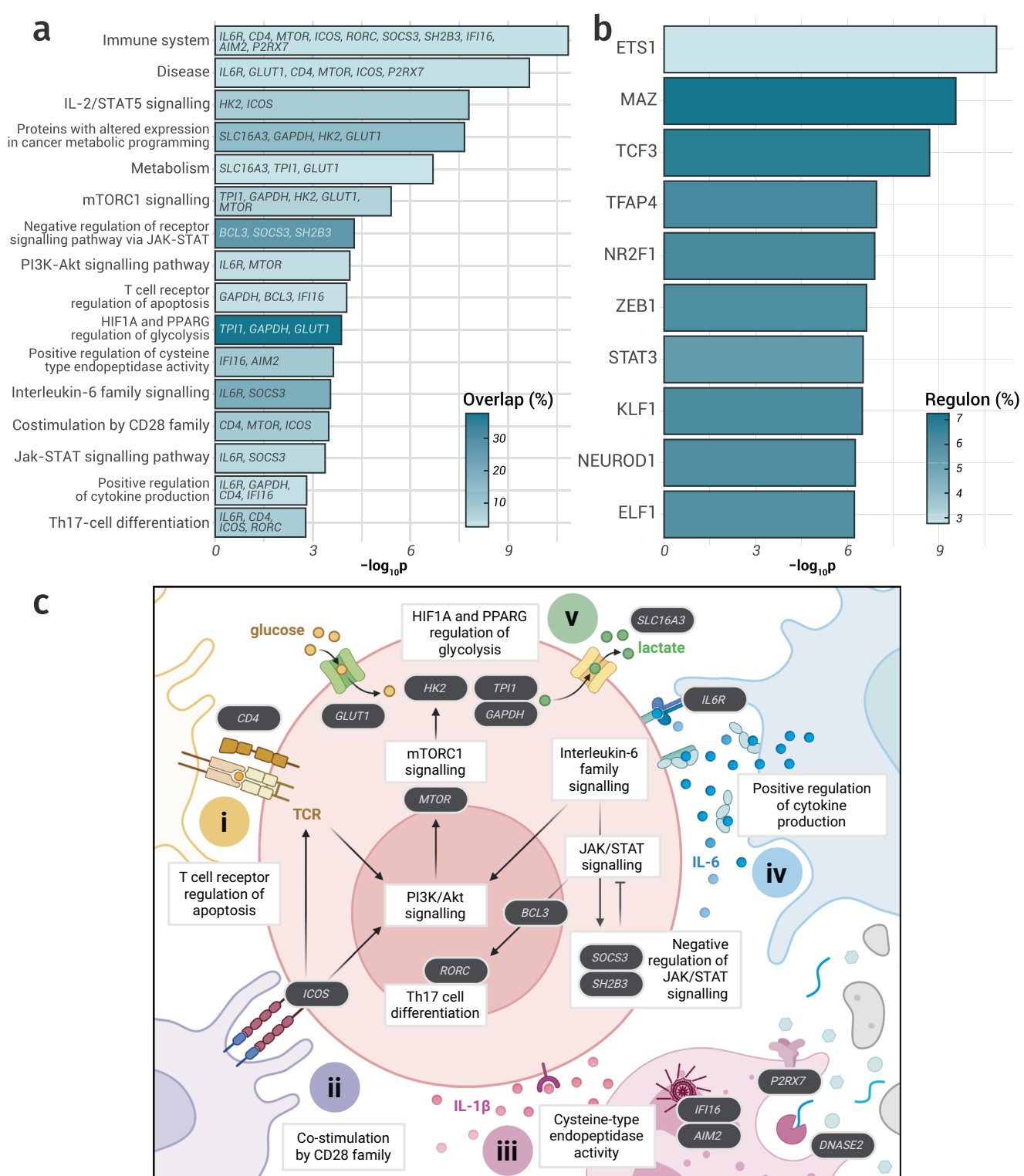

**Fig. 7 | Enrichment analyses of genes linked to IL-6–associated DNAm. a** Top five enriched gene sets, plus eleven enriched terms with specific relevance to Th17 activation and IL-6 signalling. Bars are colored by the percentage of overlapping genes; genes mapped onto the aetiological cascade from T-cell activation to glycolysis-fuelled Th17 proliferation are indicated. **b** Top ten transcription factors (TFs) whose regulons are most enriched in the genes linked to IL-6-associated DNAm using DoRothEA TF target annotations. **c** Selected enriched terms (white boxes) and mapped genes (black boxes) arranged along the pathway to Th17 differentiation, including four activation signals: (i) TCR activation, (ii) co-stimulation, (iii) IL-1β stimulation, (iv) IL-6/JAK/STAT3 signalling, culminating in (v) mTORC1-driven glycolysis fuelling rapid proliferation and effector functions. Created in BioRender. *Data underlying Fig. 7 is provided in* Supplementary Data 11 and 12.

and ELF1, both key regulators of immune cell activation[43,44], and TCF3, a central factor in B-cell commitment and early T-cell development[45]. Collectively, these results suggested that IL-6-associated DNAm was concentrated within regulatory networks that govern leukocyte differentiation and cytokine signalling, whilst also highlighting a distinction between the TFs whose regulons are enriched in genes connected to IL-6-associated DNAm and those with binding sites located near the CpGs themselves.

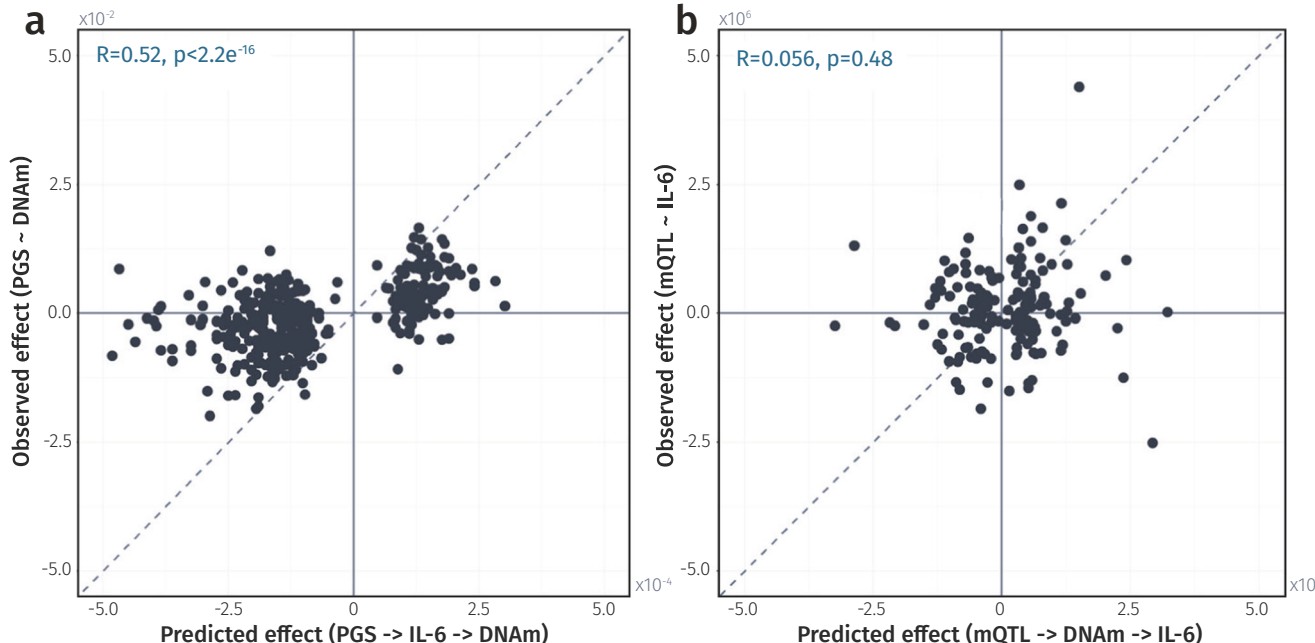

**Fig. 8 | Triangulation analysis of the directional relationship between IL-6 and DNAm. a** Scatter plot comparing the observed effect (PGS-DNAm associations) with the predicted effect (combining PGS effect on IL-6 and IL-6 CpG associations). Correlation coefficients (R) and associated *p*-values are shown in blue.

**b** Scatter plot comparing the observed (mQTL IL-6 associations) and predicted effects (combined effect of SNP on DNAm, and DNAm on IL-6). Each point represents an IL-6-associated CpG. *Data underlying* Fig. 8 *is provided in* Supplementary Data 13 and 14.

To clarify how the enriched pathways converged on immune-cell activation and IL-6 biology, we assembled a mechanistic map tracing terms from T-cell receptor (TCR) and co-stimulatory inputs through to mTORC1-driven glycolysis and expansion of pro-inflammatory effectors. Within this axis, Th17 cells represented a key conduit linking IL-6 signalling to the IL-17-dominated inflammation characteristic of many chronic inflammatory diseases[46]. We positioned eleven enriched terms along this aetiological trajectory and highlighted 17 genes with specific relevance for this process (Fig. 7c). These encompassed four co-ordinated cues critical for Th17 activation: TCR engagement (Fig. 7ci), co-stimulatory inputs (Fig. 7cii), and cytokine signals, including IL-1β (Fig. 7ciii) and IL-6 (Fig. 7civ)[47]. The latter of these triggers JAK/STAT3 activation, upregulating *RORC*, the lineage-defining TF of Th17 cells[48]. These inputs converge on mTORC1, promoting glycolysis as activated cells metabolically shift to rapid ATP generation and anabolic flux (Fig. 7cv)[49]. Consistent with this model, our gene set included canonical glycolytic genes (*GLUT1*, *HK2*, *TPI1*, *GAPDH*, and the lactate exporter *SLC16A3*), which support Th17 persistence in inflamed, often hypoxic, tissues, and genes relevant for all four arms of T-cell activation[50]. The prominence of these pathways in our results suggested that identified genes captured mechanisms that connect IL-6 signalling to heightened inflammatory risk and disease pathogenesis.

**Triangulation supports IL-6 as a driver of DNAm at the majority of CpGs**

Overall, the genes linked to IL-6-associated methylation played a vital role in immune and inflammatory processes. However, ascertaining the directionality of these relationships was far from straightforward. Although we were able to map genes and pathways on an aetiological cascade predominantly downstream of IL-6, immune and inflammatory responses are inherently complex. Cytokines such as IL-6 can initiate and amplify processes that activated them, creating positive feedback loops. As a result, many pathways, such as inflammasome-mediated release of IL-1β and PI3K/Akt/mTOR signalling, can occur both upstream and downstream of IL-6 production[51–54].

To disentangle these interactions, we employed triangulation analysis, using genetic variants as proxies for IL-6 levels and DNAm. This framework assumes that if genetically-determined outcome levels ("observed effects") are predominantly driven by an exposure, then they should be predictable from a combination of genetically-determined exposure and exposure-outcome association ("predicted effects")[13,19,24]. The correlation between observed and predicted effects then quantifies the combined support for a given direction across CpGs, even if there is insufficient power at individual sites. By performing this analysis bidirectionally, we comparatively inferred whether the data more strongly supported IL-6 as a cause or consequence of DNAm[55].

Building on conclusions from previous EWAS, where blood-based DNAm was often identified as a marker of complex traits, and informed by our gene set mapping, we first examined whether IL-6 might be driving DNAm[13,24,56]. Correlations between predicted (through IL-6) and observed effects provided strong support for this direction of effect ($R = 0.52$, $p < 2.2 \times 10^{-16}$, Fig. 8a and Supplementary Data 13). In contrast, testing the reverse pathway yielded only a weak and non-significant correlation ($R = 0.06$, $p = 0.48$, Fig. 8b and Supplementary Data 14), indicating that IL-6 plausibly drives methylation at the majority of IL-6-associated CpGs.

**Causal inference supports DNAm at *SOCS3* as mediating IL-6 effects on inflammatory risk and disease**

Considering the finding that the majority of IL-6-associated CpGs likely reflected a response to IL-6 levels, we next sought to identify candidate loci where this effect was most strongly supported. We leveraged publicly available transcriptomic data from isolated immune cells stimulated with IL-6 in vitro, evaluating which of the 421 genes were transcriptionally responsive[57]. This analysis uncovered 52 genes, including core inflammasome components *AIM2* ($\Delta = 4.70$, $p_{fdr} = 0.017$) and *IFI16* ($\Delta = 3.15$, $p_{fdr} = 0.017$), as well as *SOCS3* ($\Delta = 22.44$, $p_{fdr} = 0.020$; Fig. 9a and Supplementary Data 15).

To investigate whether DNAm at these IL-6 responsive loci mediated IL-6 effects on nine traits and disease, we performed mediation analysis. The tested phenotypes covered major risk factors, such as CRP, BMI, and lipid

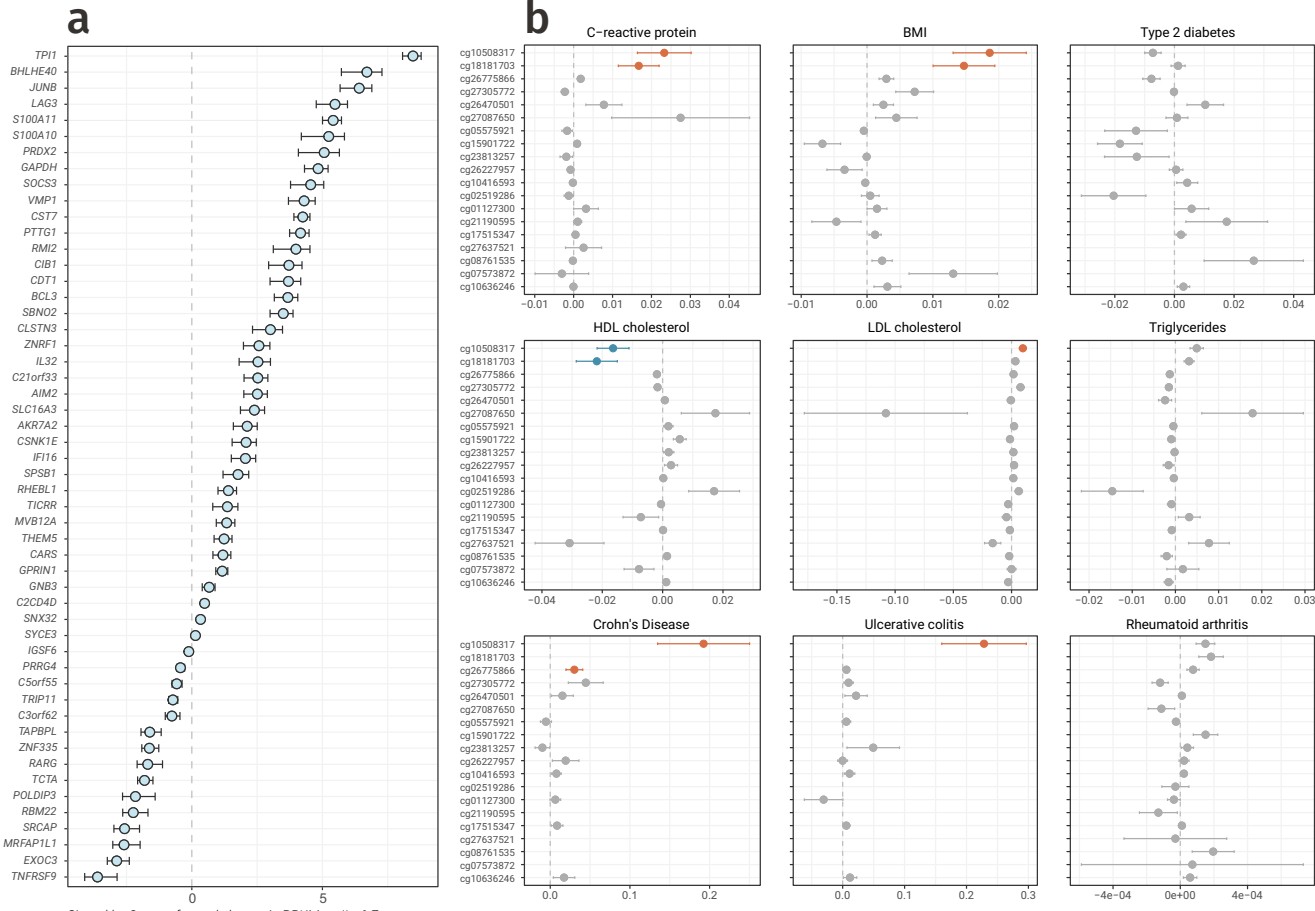

**Fig. 9 | Downstream effects of IL-6. a** Forest plot showing the 52 genes linked to IL-6-associated DNAm that were responsive to IL-6 stimulation in isolated immune cells in vitro. **b** Forest plots of CpGs ranked in the top ten for at least one tested trait in the mediation analysis. Bars are coloured by the direction and significance of the mediated effect: orange for significant positive mediation, blue for significant negative mediation, and grey for non-significant findings after correcting for multiple testing. *Data underlying* Fig. 7 *is provided in* Supplementary Data 15 and 16.

levels (triglyceride, HDL and LDL cholesterol levels), as well as a spectrum of IL-6-associated diseases, including T2D, RA, and two forms of IBD (Crohn's disease and ulcerative colitis). Two CpGs connected to *SOCS3* expression plausibly propagated IL-6 increases in CRP and BMI, as well as decreases in HDL cholesterol. One of these CpGs also showed evidence for mediating IL-6 increases in LDL cholesterol and risk of both forms of IBD ($p_{fdr} < 0.05$). Collectively, this experimental and causal evidence not only reinforced conclusions from triangulation that IL-6-associated CpGs are predominantly responsive to IL-6, but also highlighted *SOCS3* as a critical regulatory locus that may drive IL-6 effects on inflammatory health and disease (Fig. 9b and Supplementary Data 16).

### Mendelian randomisation reveals a CpG plausibly influencing IL-6 production amid predominantly downstream effects

Although triangulation pointed to DNAm predominantly reflecting a response to IL-6, we also evaluated causal effects at the individual CpG level. To this end, we applied bidirectional 2SMR, an instrumental variable approach that uses genetic proxies to assess if DNAm at specific CpGs might be regulating or responding to IL-6. Site-specific analysis such as these are inherently vulnerable to low power, which can bias estimates towards the null, and as most CpGs were instrumented by only a few independent mQTLs and since IL-6 GWAS have only identified a small number of independent loci, this was also true here. Despite this limitation, 2SMR identified a CpG (cg26663590) where DNAm may causally influence IL-6 production ($\beta = 0.11$, $p_{fdr} = 3.6 \times 10^{-2}$; Supplementary Data 17 and 18), representing a notable exception to the overall pattern observed through

triangulation and mediation. This CpG was instrumented by two independent single nucleotide polymorphisms (SNPs) (rs7185941 and rs7186573), with the causal signal primarily driven by rs7186573 ($p_{rs7186573} = 7.5 \times 10^{-4}$ vs. $p_{rs7185941} = 1.2 \times 10^{-1}$).

DNAm at this CpG was correlated with Nuclear Factor of Activated T cells 2 Interacting Protein (*NFATC2IP*) expression in BIOS ($\beta = -0.005$, $p_{fdr} = 9.3 \times 10^{-4}$) but shared underlying genetic architecture with *ATP2A1*. *NFATC2IP* is an established regulator of NFAT-driven transcription and has been linked to *IL6* expression[58–60]. While *ATP2A1* is best characterised in muscle, its role in Ca2+ transport is relevant to NFAT TFs, which are critically sensitive to intracellular calcium dynamics[61,62]. Therefore, these findings suggested that this putatively causal CpG may influence IL-6 through modulation of NFAT TF activity[63].

To validate the connections between this CpG and expression, we examined publicly available data containing matched methylation and expression from B-cells at various stages of development (GSM1104678; Supplementary Fig. 3a, b), and CD4 + T-cells (GSM4364456; Supplementary Fig. 3c, d)[27,28]. Across both datasets, this CpG was associated with *NFATC2IP* and *ATP2A1* (all $p_{fdr} < 0.1$). For *NFATC2IP*, the direction was consistent across BIOS and both immune cell types, with higher DNAm correlated with lower expression (BIOS: $\beta = -0.005$, $p_{fdr} = 9.3 \times 10^{-4}$; CD4 + T-cells: $R = -0.44$, $p = 0.085$; B-cells: $R = -0.64$, $p = 0.002$). In contrast, the relationship between *ATP2A1* differed between cell-types, showing a positive correlation in CD4 + T-cells ($R = 0.65$, $p = 0.007$) but a negative one in B-cells ($R = -0.49$, $p = 0.023$). Despite differences, results illustrated that this CpG plausibly regulated transcription in whole blood

and isolated immune cells and could therefore act upstream of IL-6 production.

### cg26663590 may influence immunometabolic risk as well as IL-6 production

In addition to its potential role as a driver of IL-6 in this study, DNAm at cg26663590 has also been implicated in BMI, incident T2D, and Crohn's disease[13,29,32,64]. Coupled with biological and functional evidence linking cg26663590 to altered expression of NFAT TF regulators, this motivated investigation into the clinical significance of this CpG[65]. To this end, we evaluated causal relationships between cg26663590 and the nine traits assessed in our prior mediation analysis, this time using a bidirectional 2SMR framework. For all but one of the tested traits (88.9%), including T2D and both forms of IBD, evidence supported cg26663590 as a causal ($p_{fdr} < 0.05$; Supplementary Data 19) rather than a consequential factor ($p_{fdr} \geq 0.05$; Supplementary Data 20). The predicted effect was negative for HDL cholesterol ($\beta = -0.027$, $p_{fdr} = 2.1 \times 10^{-3}$) and positive for CRP, BMI, and all tested diseases, in line with a protective effect from decreased DNAm at this CpG. This conclusion was further supported by the positive association between cg26663590 and IL-6 ($\beta = 0.004$, $p_{fdr} = 1.5 \times 10^{-2}$). In summary, this CpG represented a site worth investigating for plausibly driving immunometabolic disease risk through IL-6-dependent mechanisms.

### Discussion

In this study, we performed an epigenome-wide analysis of the relationship between circulating IL-6 levels and DNAm in blood from 4361 individuals across three cohorts. We identified 401 IL-6-associated CpGs, enriched for open chromatin across multiple tissues and immune cell lineages. These sites were located near TFBS relevant to IL-6 biology, including the primary transcriptional regulator of *IL6*, NF-κB[66], as well as mediators of unfolded protein (Atf4, CHOP) and oxidative stress responses (Nrf2)[33–36]. Integrative analyses linked these CpGs to inflammatory genes central to four interconnected arms of Th17-cell differentiation: TCR signalling (*CD4, LAG3, TAPBPL*), co-stimulation (*ICOS, TNFRSF8, TNFRSF9*), inflammasome activation and IL-1β secretion (*P2RX7, AIM2, IFI16*), and IL-6/JAK/STAT pathways (*IL6R, BCL3, SOCS3*). In addition, CpGs were connected to *RORC*, a gene encoding the Th17 lineage defining TF RORγt, as well as canonical genes for the mTORC1-driven metabolic shift to glycolysis that supports proliferative and pro-inflammatory effector functions (*HK2, GLUT1, MTOR*). Over-representation analysis supported these findings, identifying highly specific pathways including Th17 cell differentiation. Collectively, our findings underscore the contribution of epigenetic mechanisms to the regulation of critical inflammatory and metabolic processes, which are either marked, stabilised, or controlled by DNAm[67].

To investigate whether DNAm was the cause or consequence of IL-6, we applied three complementary causal inference approaches. Bidirectional triangulation, which takes a holistic view of causality and jointly evaluates evidence for directions across CpGs, indicated that DNAm was predominantly responding to rather than regulating IL-6. This pattern mirrored findings from previous EWAS of BMI, Crohn's, and CRP[13,24,32], as well as prior 2SMR and experimental studies[6,56]. Consistent with this downstream relationship, 52 genes connected to IL-6-associated CpGs were responsive to IL-6 stimulation in publicly available in vitro data from isolated immune cells. To assess whether DNAm at these loci might mediate the effects of IL-6 on inflammatory risk and disease, we performed a mediation analysis. This identified two CpGs, both annotated to *SOCS3*, that plausibly mediated IL-6-driven increases in BMI and CRP and decreases in HDL cholesterol. One of these CpGs additionally mediated IL-6 increases in LDL cholesterol and both forms of IBD (Crohn's disease and ulcerative colitis). Taken together, these analyses highlighted epigenetic regulation of *SOCS3* expression as plausibly playing a role in propagating the adverse effects of elevated IL-6.

By contrast, bidirectional 2SMR revealed a putatively causal association in the reverse direction: DNAm at a CpG previously implicated in BMI and

T2D, may also influence IL-6 production. In both large-scale blood-based cohorts and in vitro data from isolated B- and CD4 + T-cells, methylation at this site was inversely correlated with the expression of *NFATC2IP*, which regulates NFAT-driven cytokine transcription. Therefore, we propose that this CpG may affect IL-6 through modulation of NFAT-dependent signalling[65]. Further analyses revealed that this site may have broader effects, potentially driving not only IL-6 but also a spectrum of immunometabolic risk factors and diseases, including BMI, T2D, and two forms of IBD, suggestive that epigenetic regulation may integrate into the multilayered network controlling IL-6 production in immune cells.

This study comprehensively characterised and mapped IL-6-associated DNAm, accounting for key confounders including smoking and twelve distinct immune cell subtypes. By performing EWAS on both IL-6 and CRP, we directly compared attenuation of effects when adjusting for the other inflammatory mediator. These analyses revealed that CRP-associated methylation was more sensitive to IL-6 adjustment than vice versa, suggesting that IL-6 lies upstream of CRP in the causal pathway. This aligns with established biology, in which IL-6 drives hepatic CRP production as part of the acute phase response[21].

Previous studies have also investigated links between DNAm and IL-6, providing important context for our findings. In a trial of breast cancer patients, chemotherapy triggered hypomethylation at eight CpGs in parallel with increased IL-6 levels, suggesting a role for DNAm in chemotherapy-driven inflammation[68]. We replicated six of these CpGs (75%), although one was excluded from our final set of CpGs due to sensitivity to immune cell-type composition. While this manuscript was under review, a separate IL-6 EWAS also replicated four of these chemotherapy-associated CpGs, meaning that these sites that have now been independently observed in three studies[66]. This same EWAS reported 178 IL-6–associated CpGs, of which we replicated 66 (37.1%). However, 18 of these 66 sites (27.3%) were excluded in our final results due to confounding from extended cell counts, underscoring the critical importance of accurately modelling leukocyte heterogeneity in blood-based EWAS. In this study, IL-6-associated CpGs were further characterised through systematic follow-up analyses, which linked them to core inflammatory processes and assessed their directions and disease relevance. Moreover, a prior study developed an IL-6 predictor comprising 35 CpGs[69], of which two were identified here, while the remaining 33 have yet to be replicated in any independent study.

One notable finding was that STAT3, which directly responds to IL-6 signalling, did not have binding sites enriched within 50 bp of IL-6-associated CpGs, but that nevertheless, genes linked to these CpGs were enriched in the STAT3 regulon. This could reflect IL-6 modulation of STAT3 targets through multiple mechanisms, both directly through STAT3 phosphorylation, but also indirectly through DNAm modulation of other TFBS, such as NF-κB, which cooperate with STAT3 to regulate target genes. This interpretation would align with the CpG-poor nature of STAT3 motifs[70]. Alternatively, IL-6 could influence DNAm via STAT3-independent TFs that share downstream targets with STAT3, or via modulation of methylation enzymes, a mechanism supported by prior experimental evidence[6].

There were several key limitations to this study. Firstly, the immune response is governed by a complex network of inflammatory proteins, interacting through positive and negative feedback loops. Our analysis, however, was restricted to only two mediators: IL-6, the primary focus, and CRP, which was included as a sensitivity analysis. Secondly, although IL-6 is produced by and acts upon circulating cells, many of its downstream effects occur in other tissues, such as the liver or pancreas. Because our study relied on blood-based DNAm, it is probable that distinct directional effects would be observed in other IL-6-responsive tissues. Blood is also highly heterogeneous. While we adjusted for twelve extended immune cell-types, incorporated cell-type specific reference epigenomes, and validated findings using isolated immune cell datasets, these approaches were insufficient to pinpoint specific subtypes driving associations. In addition, multiple methods exist to estimate immune cells, all based on the same reference matrix from purified cells of between three and six donors per cell type[20,71].

Adjusting for extended cell-type proportions slightly reduced the number of IL-6-associated CpGs in this study, and there was strong concordance in effect estimates before and after this adjustment ($R = 0.96$), suggesting that our findings were largely robust to finer-grained modeling of immune heterogeneity. Nevertheless, some initial associations may have been masked by the lower resolution adjustment in base models and could be revealed in future studies using higher resolution deconvolution. Our extended characterisation of genes linked to IL-6-associated CpGs also identified several cell-type specific processes, including cytokine production by DCs, inflammasome activation in neutrophils, and metabolic reprogramming of T-cells, underscoring the importance of isolated cell-types in IL-6 biology.

Our causal inference was constrained by weak instruments, composed of only a small number of SNPs. As a result, rigorous testing of the relevance assumption was not feasible and we cannot rule out horizontal pleiotropy as an explanation for our directional findings[72]. Furthermore, the three cohorts in this study differed in population demographics, fasting status, and assay sensitivity and types. These factors likely contributed to higher heterogeneity. Since the directions of effect were largely consistent across cohorts, with fewer than 1% of IL-6-associated CpGs showing discordance ($n = 4$), we adopted a higher heterogeneity threshold, retaining more CpGs while acknowledging that this may come at the expense of precision. In addition, resulting CpGs were thoroughly characterised using external datasets, multiple causal inference approaches, and experimental data meaning that conclusions did not depend solely on discovery cohorts. Lastly, this study provided only a cross-sectional snapshot of DNAm and IL-6, and therefore cannot distinguish between acute and chronic inflammation. To confirm and extend on the links between DNAm and IL-6, future work could incorporate intervention studies, experimental approaches, or longitudinal designs that can provide stronger causal evidence.

In summary, our findings illustrated the value of integrative analytical frameworks and bidirectional causal inference for dissecting the regulatory interface between epigenetics and inflammation. IL-6 represents a critical circulating cytokine with broad implications for the initiation and progression of disease, and here we have positioned DNAm as a potential contributor to both its regulation and downstream effects on human health.

## Methods

### Cohort descriptions

**Leiden Longevity Study (LLS).** The LLS[73] is a cohort of long-lived, Dutch, Caucasian siblings ($n = 944$), who were recruited with their offspring ($n = 1671$) and their offspring's partners ($n = 744$). Between 2002 and 2006, research nurses collected non-fasted blood samples from living subjects for isolation of DNA, RNA, serum, and plasma, and these were stored at $-80\,°C$ until analysis. From these samples, IL-6 levels were assessed in plasma using a PeliKine Compact Human IL-6 ELISA kit (Sanquin Reagents, Amsterdam, The Netherlands) and plasma hsCRP was assayed using an automated enzymatic colorimetric method with a Modular P analyzer (Roche, Almere, The Netherlands).

DNAm data of whole blood samples was generated for 821 unrelated participants from the offspring-partner generation of the LLS by the Human Genotyping facility (HuGe-F, Erasmus MC, Rotterdam, The Netherlands) within the Biobank-Based Integrative Omics Studies (BIOS) consortium, funded by BBMRI-NL, a research infrastructure financed by the Dutch government (NOW 184.021.007)[74]. Genomic DNA (500 ng) was bisulfite converted using the Zymo EZ-96 DNA methylation kit (Zymo Research Corp, Irvine, CA, USA). 4 µl was then hybridized on the Infinium HumanMethylation450 BeadChip array (Illumina Inc., San Diego, CA, USA) according to the manufacturer's protocol. Data preprocessing and quality control followed the DNAmArray workflow[75] and is described in detail in Supplementary Methods (Cohort Descriptions).

Written informed consent for DNA collection and its use for genetic analyses was obtained from all participants prior to their enrolment into the study. Good clinical practice guidelines were maintained, and the study protocol was approved by the local Medical Ethical Committee of the Leiden University Medical Center. All ethical regulations relevant to human research participants were followed.

**Kooperative Gesundheitsforschung in der Region Augsburg (KORA).** The KORA F4 study (2006–2008) followed up individuals ($n = 3080$) living in the region of Augsburg, Southern Germany, aged between 32 and 81 years old[76]. Blood samples were drawn after an 8 hour fast and frozen at $-80\,°C$ until analysis. Serum concentrations of IL-6 were measured with the Human IL-6 Quantikine HS ELISA kit (R&D Systems, Wiesbaden, Germany), and hsCRP was determined by a high-sensitivity latex-enhanced nephelometric assay on a BN II analyzer (Dade Behring)[77].

Following isolation according to standard procedures, genomic DNA (750 ng) from 1707 whole blood samples was bisulfite converted with the Zymo EZ-96 DNA Methylation Kit (Zymo Research Corp, Irvine, CA, USA). 4 µl from each sample underwent amplification, enzymatic fragmentation, and application to the Infinium HumanMethylation450 BeadChip array (Illumina Inc., San Diego, CA, USA) according to the manufacturer's protocol. The methylation module of GenomeStudio was used to extract and process the raw image data, and initial quality assessment was conducted using the Control Dashboard. For data preprocessing, a common pipeline was used and details of this are available in the Supplementary Methods (Cohort Descriptions)[78].

In accordance with the Declaration of Helsinki, written informed consent was obtained from all participants prior to their enrolment into the study. Good clinical practice guidelines were maintained, and the study protocol was approved by the Ethics Committee of the Bavarian Medical Association. All ethical regulations relevant to human research participants were followed.

**Netherlands twin register (NTR).** The NTR, a national register established in 1987, recruited Dutch twins and their families, collecting fasted whole blood samples during a home visit[79,80]. EDTA plasma was harvested and aliquoted (0.5 ml), snap-frozen in dry ice, and stored at $-30\,°C$. IL-6 levels were determined with an UltraSensitive ELISA (R&D Systems, Minneapolis, USA, Quantikine HS HSTA00C), and hsCRP was measured with the Immulite 1000 CRP assay (Diagnostic Product Corporation, USA).

DNAm data of whole blood samples was generated by the Human Genotyping facility (HuGe-F, Erasmus MC, Rotterdam, The Netherlands) within the BIOS consortium[74]. Quality control and normalization have been described in detail previously[81]. The methylation data were normalized with functional normalization, expressed as $\beta$-values, and any data points further than 3 interquartile ranges (IQR) from the nearest quartile were excluded[82]. Further details are provided in the Supplementary Methods (Cohort Descriptions).

Informed consent was obtained from all participants. The study was approved by the Central Ethics Committee on Research Involving Human Subjects of the VU University Medical Centre, Amsterdam, an Institutional Review Board certified by the U.S. Office of Human Research Protections (IRB number IRB00002991 under Federal-wide Assurance-FWA00017598; IRB/institute codes, NTR 03-180). All ethical regulations relevant to human research participants were followed.

### Cohort-based analyses

**Base models.** Cohorts were contacted to participate in the study and those with required data types were recruited. All cohorts followed a common analysis plan and all samples analysed were taken from distinct individuals (i.e., there were no repeat measurements included in the analysis). Any IL-6 measurements below the limit of detection and outlying values for both DNAm and IL-6 (more than three IQR from the nearest quartile) were removed prior to analysis rather than imputed to avoid introducing bias or heterogeneity due to the differences in assay detection limits or types between cohorts.

For each of $j$ CpGs measured in $i$ individuals, a linear regression model (see Eq. (1) for general specification) was fitted of DNAm $\beta$-values on IL-6 values, measured in pg/mL and natural log-transformed to improve Normality. $\beta$-values were used to improve downstream interpretation and harmonization with previous analyses. All models were adjusted for age (in years), sex, cell-type proportions predicted from DNAm data using the IDOL algorithm (monocytes, CD8 + T, CD4 + T, natural killer, and B-cells)[20], and technical covariates (left to the analyst's discretion). All participating cohorts are European and therefore adjustment for race or ethnicity was not relevant for this study. Cohort-specific models are described in detail in the Supplementary Methods.

$$DNAm_{\beta_{ij}} = \beta_0 + \beta_1 \cdot \ln(IL6)_i + \beta_2 \cdot age_i + \beta_3 \cdot sexFemale_i + \beta_4 \cdot CD8T_i$$
$$+ \beta_5 \cdot CD4T_i + \beta_6 \cdot NK_i + \beta_7 \cdot Mono_i + \beta_8 \cdot technical\_factors_i \quad (1)$$

**Sensitivity analyses.** Each cohort ran additional analyses, adding trichotomous smoking to the cohort-specific base models as a categorical fixed effect. In some cases, this resulted in a reduction of the sample size as there were missing values in the smoking data.

To investigate the effect of adjusting for additional cell-types whose proportions could be estimated from DNAm data using the IDOL extended algorithm, basophils, memory B-cells, naïve B-cells, CD4+ memory T-cells, CD4+ naïve T-cells, CD8+ memory T-cells, CD8+ naïve T-cells, eosinophils, monocytes, natural killer cells, and regulatory T-cells were added to the base model for all cohorts[20]. Neutrophils were excluded to avoid collinearity (as proportions for all cells sum to 1, meaning a model without neutrophils already contains information about neutrophil proportions and their addition would introduce multicollinearity)[83]. Plots comparing effect sizes from the base model with each of the sensitivity analyses were inspected and tests of correlation were performed. IL-6-associated CpGs no longer associated after adjustment were removed prior to downstream analysis ($p_{fdr} \geq 0.05$).

**Additional phenotypes.** To distinguish inflammatory signals, cohorts additionally adjusted the base IL-6 model for hsCRP (measured in mg/L). All cohorts additionally ran an hsCRP EWAS with the same specification as the IL-6 base model (see Eq. (2) for general specification) both with and without adjustment for IL-6 levels.

$$DNAm_{\beta_j} = \beta_0 + \beta_1 \cdot \ln(CRP)_i + \beta_2 \cdot age_i + \beta_3 \cdot sexFemale_i$$
$$+ \beta_4 \cdot CD8T_i + \beta_5 \cdot CD4T_i + \beta_6 \cdot NK_i + \beta_7 \cdot Mono_i \quad (2)$$
$$+ \beta_8 \cdot plate1_i + \ldots + \beta_{17} \cdot plate10_i + \beta_{18} \cdot arrayRow_i$$

Further details of all main and additional analyses, including cohort-specific model specifications, are included in the Supplementary Methods.

**Combined cohort analyses**
**Differentially methylated probes.** Cohorts provided summary statistics for each CpG, including the mean and standard error (SE) of DNAm levels, size of association with IL-6, along with the SE, nominal $p$-value, and number of individuals used to derive the IL-6 association. Results from each cohort were inspected and CpGs were removed if they were estimated from fewer than 50 participants. Probes located on sex chromosomes, in ENCODE Blacklist regions, that contain known common genetic variants, or which were ambiguously mapped were also removed[84,85]. To ensure good quality data, we inspected QQ, volcano, and Manhattan plots, alongside boxplots of the effect sizes and SE distributions across cohorts. Following these steps, data was available on 412,226 CpGs for the base IL-6 model.

The Bioconductor package bacon was used to inspect and adjust for bias and inflation of the test statistics, using default priors ($\alpha = 1.28$, $\beta = 0.36$)[86]. After running bacon, inflation and bias were re-estimated for all models to ensure that observed bias and inflation had been adequately

corrected for. Bacon-adjusted effect sizes and SEs were used as input in a fixed-effects meta-analysis in METAL[87]. Separate analyses were performed for the base model, each extended model (adjusted for smoking, extended cell counts, and hsCRP), and hsCRP models (base and IL-6 adjusted). Any CpGs where there was evidence of high heterogeneity of effect sizes between cohorts ($I^2 \geq 90\%$) were removed.

**Differentially methylated regions.** To assess the number of distinct genomic loci associated with circulating IL-6 levels, differentially methylated regions (DMRs) were identified using the DMRfinder algorithm as implemented in the DNAmArray workflow[75,88]. DMRs were defined as regions with at least three differentially methylated positions (DMPs) with an inter-CpG distance of less than 1 kb, allowing a maximum of three non-DMPs across a DMR. Then, the number of distinct loci were calculated as the total number of DMPs minus the number of DMPs in DMRs plus the number of DMRs called by DMRfinder.

**Functional annotation**
**Epigenome-wide association study (EWAS) enrichment.** Using summary data from the EWAS catalog and EWAS atlas, we investigated whether IL-6-associated CpGs had been previously associated with other phenotypes[89,90]. Following loading of data into R, we filtered results to include only large-scale, peer-reviewed, and relevant findings (with a PubMed ID, sample size above 500, finding over 100 CpGs at the threshold of inclusion into the respective database, performed in adults, reporting nominal $p$-values, and using whole blood or leukocyte samples). We also recoded traits to ensure consistency between names, for example, by combining EWAS of "BMI" and "body mass index" (see Supplementary Data 5 for full details). Finally, we combined data with summary statistics from a recent hsCRP EWAS, which was not included in either catalog[22]. This resulted in a list of 57 traits, for which we tested enrichment of associations with our CpGs using logistic regression.

**Chromatin state enrichment.** IL-6-associated CpGs were annotated to chromatin state using the Roadmap reference epigenome for peripheral blood mononuclear cells (PBMCs; E062), which includes histone modification ChIP peaks (H3K4me1, H3K4me3, and H3K27ac), marks of open chromatin, and DNase I hypersensitivity sites[30]. We mapped each probe tested on the Illumina 450 k array background to eight active and seven repressive chromatin states, recording overlap at each probe as a binary variable. To determine whether enrichment occurred more often than expected by chance, we performed logistic regression using the glm function in R to calculate and test odds ratios (ORs) of significance for each state. Nominal $p$-values were adjusted for multiple testing using the false discovery rate (FDR) and enrichments or depletions were assessed at a 5% significance threshold.

**Transcription factor binding site (TFBS) enrichment.** A 50 bp window around IL-6-associated CpGs was scanned using findMotifsGenome.pl from HOMER to evaluate enrichment of known motifs compared to a random background matched for GC content[91]. ENCODE TFBS annotation for 171 TFs and CpGs on the 450 k array was used to further investigate the size of binding sites and distance from CpG to summit[85]. TFs associated with enriched TFBS were examined for links with inflammation and, specifically, IL-6 pathways and interactions.

**Expression quantitative trait methylation (eQTM) analyses.** Blood-based RNA-sequencing and DNAm data from matched samples were available from the BIOS consortium ($n = 3152$), comprising six Dutch biobanks: namely the Cohort on Diabetes and Atherosclerosis Maastricht[92], LifeLines[93], LLS[73], NTR[79,80], Rotterdam Study[94], and Prospective ALS Study Netherlands[95]. After filtering out non-autosomal and lowly expressed genes, count data was transformed into $\log_2$ counts per million ($\log_2$CPM), and values for each gene were rank-inverse normal (RIN) transformed prior to analysis. Genomic locations of human

transcripts, exons, coding sequences, and genes were imported from the Ensembl database using makeTxDbFromEnsembl from the GenomicFeatures Bioconductor package[96]. These were used to identify all genes within 100 kb of each IL-6-associated CpG. To examine correlations between DNAm and gene expression, we carried out linear regression with RIN-transformed gene expression values as the response variable and methylation β-values as the independent variable, adjusting for the effects of age, sex, technical covariates (row, plate, and flowcell), and twelve predicted blood cell-types predicted from DNAm data.

**Colocalisation analysis.** To investigate if DNAm may be driving local gene expression, we performed colocalisation analysis, exploring if these two molecular traits shared underlying causal variants[97]. Since the assumption of single causal variants in traditional colocalisation is a spurious one, we used the Sum of Single Effects (SuSiE) regression framework in combination with coloc[98]. SuSiE improves fine-mapping and colocalisation by modelling multiple causal variants within a region simultaneously, rather than assuming a single causal signal. It separates the statistical evidence supporting each variant by conditioning on other signals, thereby reducing confounding between correlated variants and yielding more reliable colocalisation inferences[99].

To perform this analysis, we downloaded mQTL data from the Genetics of DNA Methylation Consortium (GoDMC), saving associations between DNAm at the IL-6 associated CpGs and all SNPs within ±1000 kb regardless of significance[40]. For each CpG-gene pair, we used the full eQTL summary statistics from eQTLGen and stored associations between the identified SNPs and expression of nearby genes alongside minor allele frequencies[39]. Linkage disequilibrium (LD) matrices based on reference haplotype data from the publicly released Phase 3 (Version 5) 1000 Genomes Project population were provided by the LDlinkR package, which allows access to the LDlink API from the R console[100]. Variances in methylation and expression at each IL-6-associated CpG and linked gene were calculated using available DNAm and RNA-seq data from the BIOS consortium. Finally, the coloc.susie function from the coloc R package was used to run the analysis with default priors for the probability a SNP was associated with DNAm ($p_1 = 1.0 \times 10^{-4}$), the probability a SNP was associated with expression ($p_2 = 1.0 \times 10^{-4}$), and the probability a SNP was associated with both traits ($p_{12} = 5.0 \times 10^{-6}$). Colocalisation of DNAm and expression signals was considered confirmed if the posterior probability of H4, the hypothesis where both gene expression and DNAm share causal variants, was above 0.9.

**Over-representation analysis.** Following integrative analyses, we generated a list of plausible target genes for each IL-6-associated CpG. The associated gene names were then used as input for over-representation analysis, using eleven recent (updated in the last 6 years) databases relating to human health and disease downloaded from Enrichr (BioPlanet 2019, Elsevier Pathway Collection, GeDiPNet 2023, GO Biological Process 2023, KEGG Human 2021, MsigDB Hallmark 2020, OMIM, PhenGenI Association 2021, PheWeb 2019, Reactome 2022, and WikiPathway Human 2021). These were imported into R and analyses were performed using the enrichr function from clusterProfiler[101]. P-values were adjusted for multiple testing using FDR and significance was assessed at the 5% level.

**TF regulon enrichment.** To identify TFs potentially regulating the set of genes connected to IL-6-associated CpGs, we performed TF enrichment analysis using the DoRothEA human regulon database. DoRothEA provides curated TF-target interactions with confidence levels ranging from A (highest) to E (lowest). We restricted this analysis to high or medium confidence interactions (levels A to D) and considered three nested subsets: A_B, A_B_C, and A_B_C_D. The input gene list consisted of unique HGNC gene symbols from our dataset, intersected with the DoRothEA target universe to define the background. For each TF, we calculated the statistical significance of the overlap between its regulon

and the input gene set using a one-sided hypergeometric test, with the universe defined as all genes present in DoRothEA. P-values were adjusted for multiple testing using FDR and significance was assessed at the 5% level.

## Causal inference

**Triangulation analyses.** To perform triangulation analyses, we interrogated the correlation between the observed effect of an instrumental variable on an outcome (i.e., methylation quantitative trait locus (mQTL)-IL-6 or polygenic score (PGS)-DNAm associations) and the predicted effect via the exposure. This analysis assumed that if the effect of an exposure on an outcome is causal, then it would be possible to predict the effect of a genetic instrument on that outcome via the combination of the instrument's effect on the exposure and the exposure's effect on the outcome.

In detail, when looking at the effect of DNAm on IL-6 levels (consequential analysis), the observed effect is the association between the top mQTL and ln(IL-6), extracted from full GWAS summary data[102]. The predicted effect can then be calculated by combining the mQTL effect extracted from GoDMC data[40] (the association effect of +1 effect allele with DNAm, $\beta_{mQTL}$) and the EWAS effect (the association effect of +1 ln(IL-6) with DNAm, $\beta_{EWAS}$). Since we wanted to predict the effect of +1 effect allele on IL-6, the calculation for the predicted effect was then $\beta_{mQTL}/\beta_{EWAS}$ in this instance. When looking in the reverse direction (i.e., IL-6 as a cause of DNAm), the observed effect is a PGS, where the effect of two IL-6-associated SNPs on DNAm were weighted by their effect allele frequencies (EAF). The predicted effect used equivalent EAF weighting and was calculated by multiplying the effect of the PGS on IL-6 ($\beta_{PGS}$) with the EWAS effect ($\beta_{EWAS}$). The observed and predicted effects in both directions were then plotted and correlation was assessed using a Pearson correlation coefficient.

**Transcriptional responses to IL-6.** To identify CpGs that were plausibly regulating transcription in response to IL-6, we used experimental data from a study quantifying transcriptomic changes induced in isolated CD4 + T-cells following IL-6 exposure (GSE65621)[57]. Data was provided as reads per kilobase of transcript, per million reads mapped (RKPM) for IL-6-exposed (n = 9) and control (n = 9) samples, following 3 days of culturing in vitro. Differential expression of 402 genes linked to IL-6 DNAm was calculated using linear models fitted in limma, adjusting for patient status and using FDR to correct for multiple testing.

**Mediation analysis.** For IL-6-associated CpGs connected to genes that responded to IL-6 stimulation in vitro, we performed a mediation analysis. This tested if DNAm mediated the causal effect of IL-6 on downstream traits. The tested phenotypes covered a number of key risk factors for diseases, such as high-density lipoprotein (HDL) cholesterol, body mass index (BMI), and CRP levels, as well as a diverse range of IL-6-associated diseases, including T2D, RA, and two types of IBD (Crohn's disease and ulcerative colitis).

For each CpG and trait, we estimated (a) the effect of IL-6 on DNAm at that CpG, (b) the effect of DNAm at that CpG on the trait, and (c) the total effect of IL-6 on the trait. Each of these effects were estimated using genetic proxies for the exposure in an instrumental variable framework, using either the Wald ratio of inverse variance weighted (IVW) estimate. Instrumental SNPs were obtained from recent large-scale GWAS, including for IL-6 (ebi-a- GCST90012005), CRP (ebi-a-GCST90029070), BMI (ieu-b-40), T2D (ebi-a-GCST006867), HDL cholesterol (ieu-b-109), LDL cholesterol (ieu-b-110), triglycerides (ieu-b-111), RA (ukb-d-M13_RHEUMA), Crohn's disease (ieu-a-12), and ulcerative colitis (ieu-a-970).

The total effect (c) was decomposed into an indirect (mediated) effect (a × b) with variance estimated using the delta method, and a direct (non-mediated) effect (c–ab). For each trait, CpG-specific mediated p-values were corrected for multiple testing using FDR, and significance was assessed at the 5% level.

**Two-sample Mendelian randomization**. To assess site-specific directional effects between IL-6 and DNAm at IL-6-associated CpGs, we used the TwoSampleMR package to perform bidirectional two-sample Mendelian randomisation (2SMR). This instrumental variable based method uses GWAS summary statistics to infer that a risk factor causally influences an outcome. 2SMR relies on several key assumptions, namely that instruments are relevant, independent, and that there is no horizontal pleiotropy. To interrogate the effects of DNAm at our CpGs on IL-6, we extracted SNP-based *cis*-mQTL data from the GoDMC and combined these with summary statistics from a large-scale GWAS of IL-6[40,102]. For some CpGs (*n* = 169, 42.14%), there was insufficient data available to interrogate the effects of DNAm at that CpG. For the remaining CpGs, between one and five independent SNPs with data on both their *cis*-association with DNAm and association with IL-6 were used as instruments. Following harmonisation of the data and clumping of the instrumental SNPs, these were combined using the Wald ratio (for single mQTL instruments) or IVW methods (for multiple, independent mQTLs).

To interrogate the influence of IL-6 on DNAm at identified CpGs, independent GWAS variants from a large-scale analysis was used. Of the two variants that could instrument IL-6, there was *trans*-mQTL data in GoDMC available for both of them (rs4537545, rs6734238). The GWAS summary statistics and mQTL effects were then combined using the IVW method and the TwoSampleMR package in R. To interrogate directional effects between DNAm at CpGs and inflammatory traits, the same GWAS and traits were used for both the mediation analysis and 2SMR analyses. For all analyses, *p*-values were adjusted for multiple testing using the FDR method and potential causal effects were assessed at the 5% significance threshold.

The following cohorts were used to derive both mQTL effects in GoDMC and IL-6 GWAS effects and therefore had overlapping individuals in both the exposure and outcome datasets for the 2SMR analysis: Lothian Birth Cohorts 1921 (GoDMC: 435 samples; IL-6 GWAS: 166 samples), Lothian Birth Cohorts 1936 (GoDMC: 905 samples; IL-6 GWAS: 759 samples), LLS (GoDMC: 718 samples; IL-6 GWAS: 1798 samples), NTR (GoDMC: 2757 samples; IL-6 GWAS: 3668 samples), Rotterdam Study (GoDMC: 1472 samples; IL-6 GWAS: 599 samples), and TwinsUK (GoDMC: 843 samples; IL-6 GWAS: 1103 samples). Therefore, overall the overlap was low considering that both meta-analyses incorporated data from over 26 cohorts.

**eQTM validation**. In order to validate eQTM effects at a CpG identified by 2SMR as causally influencing IL-6, we used publicly available DNAm and expression data from isolated CD4 + T- and B-cells[27,28]. Counts were normalised to $\log_2$CPM and DNAm was represented by $\beta$-values. Correlations between genes identified by large-scale association and colocalisation analyses and the CpG were calculated using the Pearson correlation coefficient and significance was assessed at the 5% level.

**Statistics and reproducibility**

Unless stated otherwise, all calculations were performed using R version 4.2.2. For all meta-analyses, METAL, version 2011-03-25 was used. TFBS enrichment analyses were performed using HOMER v3.1. Total sample size of this EWAS meta-analysis was 4361, but data from all individuals was not available for all CpGs. Information on exact sample sizes is available in the Supplementary Data.

**Reporting summary**

Further information on research design is available in the Nature Portfolio Reporting Summary linked to this article.

**Data availability**

The authors declare that all data supporting the findings of this study are available within the Supplementary Information and Data. Specific data underlying each Figure is noted in both the Figure Legend and in each Supplementary Data sheet. Regarding individual-level data from the cohorts involved: The informed consents given by KORA study participants does not cover data posting in public databases. However, data are available upon request from KORA Project Application Self-Service Tool (). Requests can be submitted online and are subject to approval by the KORA Board. The HumanMethylation450 BeadChip data from the NTR and LLS are available as part of the Biobank-based Integrative Omics Studies (BIOS) Consortium in the European Genome-phenome Archive (EGA), under the accession code EGAD00010000887. Additional -omic and phenotype data are available upon request via the BBMRI-NL BIOS consortium. All data can be requested by bona fide researchers from the respective cohorts. Information about the individual studies analysed in this manuscript can be found in the Supplementary Methods. All other data used in this study is publicly available: EWAS summary statistics can be downloaded from the EWAS Catalog[90] and EWAS atlas[89], reference epigenome data is available from ROADMAP[30], TFBS data is available within the HOMER software[91], full mQTL summary statistics can be requested from GoDMC[40], eQTL data can be downloaded from eQTLgen[39], IL-6 GWAS summary statistics are available upon request from the CHARGE Inflammation Working Group[102], LD matrices can be accessed using LDlink[100], variances in methylation and expression were calculated from data generated by the Biobank-based Integrative Omics Study (BIOS), and libraries for GSEA were downloaded directly from Enrichr (https://maayanlab.cloud/Enrichr/ ). Instrumental SNPs were obtained from recent large-scale GWAS, including for IL-6 (ebi-a- GCST90012005), CRP (ebi-a-GCST90029070), BMI (ieu-b-40), T2D (ebi-a-GCST006867), HDL cholesterol (ieu-b-109), LDL cholesterol (ieu-b-110), triglycerides (ieu-b-111), RA (ukb-d-M13_RHEUMA), Crohn's disease (ieu-a-12), and ulcerative colitis (ieu-a-970). Data from isolated CD4 + T-cells following IL-6 exposure can be accessed via GEO (GSE65621). Data containing matched methylation and expression from B-cells at various stages of development (GSM1104678), and CD4 + T-cells (GSM4364456) is also accessible freely from GEO.

**Code availability**

All custom code for these analyses is freely available at https://github.com/ nebulyra/il6_ewas and is stored on Zenodo (https://zenodo.org/records/ 17930604)[103]. Unless stated otherwise, all calculations were performed using R version 4.2.2. For all meta-analyses, METAL, version 2011-03-25 was used. TFBS enrichment analyses were performed using HOMER v3.1. All the software and programmes used to conduct these analyses are freely available.

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

## Acknowledgements
The authors thank the staff and participants of all cohorts involved in this study for their contributions. The work of L.S. was supported by the Joint Programming Initiative "a Healthy Diet for a Healthy Life" (JPI-HDHL) DIMENSION project [ZonMW project number: 529051021]. This paper makes use of data generated by the BIOS consortium, and access to this data can be requested from https://ega-archive.org/datasets/EGAD00010000887. Funding for the BIOS consortium was provided by the Netherlands Organization for Scientific Research (NWO 184.021.007), made available as a Rainbow Project of the Biobanking and Biomolecular Research Infrastructure Netherlands (BBMRI-NL). This paper makes use of data generated by the CHARGE Inflammation Working Group. **NTR** This research was funded by Genotype/phenotype database for behaviour genetic and genetic epidemiological studies (ZonMW Middelgroot 911-09-032); Netherlands Twin Registry Repository: researching the interplay between genome and environment (NWO-Groot 480-15-001/674); the Avera Institute, Sioux Falls (USA) and the National Institutes of Health (NIH R01 HD042157-01A1, MH081802, Grand Opportunity grants 1RC2 MH089951 and 1RC2 MH089995); NWO, project 184.034.019, and by the BBRMI-NL-financed BIOS Consortium (NWO 184.021.007); epigenetic data were generated at the Human Genomics Facility (HUGE-F) at EUR. J.v.D. is supported by NWO Large Scale infrastructures, X-Omics (184.034.019). DIB acknowledges the Royal Netherlands Academy of Science Professor Award (PAH/6635). **KORA F4** The KORA research platform (KORA, Cooperative Health Research in the Region of Augsburg) was initiated and financed by the Helmholtz Munich German Research Center for Environmental Health, which is funded by the German Federal Ministry of Education and Research and by the State of Bavaria. Furthermore, KORA research was supported within the Munich Center of Health Sciences (MC Health), Ludwig-Maximilians-Universität, as part of LMUinnovativ. The work was supported by the German Federal Ministry of Education and Research (BMBF) within the framework of the EU Joint Programming Initiative "A Healthy Diet for a Healthy Life" (DIMENSION; grant number 01EA1902A). The German Diabetes Center is supported by the Ministry of Culture and Science of the state of North Rhine-Westphalia (Düsseldorf, Germany) and the German Federal Ministry of Health (Berlin, Germany). This study was supported in part by a grant from the German Federal Ministry of Education and Research to the German Center for Diabetes Research (DZD). **LLS** The LLS was supported by a grant from the Innovation-Oriented Research Program on Genomics (SenterNovem IGE01014 and IGE05007), the Centre for Medical Systems Biology, and the National Institute for Healthy Ageing (Grant 05040202 and 05060810), in the framework of the Netherlands Genomics Initiative/Netherlands Organization for Scientific Research. Funders had no say in the study design, collection, analysis, interpretation, writing, or decision to submit this work.

## Author contributions
L.S. conceived and designed the study, performed data preprocessing and quality control, undertook cohort-specific and main analyses, interpreted results, and drafted the manuscript; B.T.H. conceived and designed the study, supervised conduct of the study, iteratively revised the manuscript, and provided critical intellectual contributions and interpretations of results; J.v.D., T.D., and R.W. performed cohort-specific analyses, provided summary statistics for the meta-analyses, provided critical intellectual contributions and interpretations of results; Y.X., J.O., and J.T.B. provided critical intellectual contributions and interpretations of results; M.B., G.W., C.G., C.H., W.K., A.P., E.C.D.G. M.W., D.I.B., and P.E.S. provided and verified data, phenotype acquisition, and harmonization for these analyses, provided critical intellectual contributions and interpretation of results. All authors read and approved the manuscript prior to submission. All collaborators of this study that have fulfilled the criteria for authorship required by Nature Portfolio journals have been included as authors, as their participation was essential for the design and implementation of the study. Roles and responsibilities were agreed among collaborators ahead of the research. This work includes findings that are locally relevant, which have been determined in collaboration with local partners. This research was not severely restricted or prohibited in the setting of the researchers, and does not result in stigmatization, incrimination, discrimination, or personal risk to participants. Local and regional research relevant to our study was considered in citations.

## Competing interests
The authors declare no competing interests.

## Additional information

[1]Molecular Epidemiology, Department of Biomedical Data Sciences, Leiden University Medical Center, Leiden, The Netherlands. [2]Department of Biological Psychology,
Vrije Universiteit Amsterdam, Amsterdam, The Netherlands. [3]Amsterdam Public Health Research Institute, Amsterdam UMC, Amsterdam, The Netherlands. [4]Research
Unit Molecular Epidemiology, Institute of Epidemiology, Helmholtz Munich, German Research Center for Environmental Health, Neuherberg, Germany. [5]Department of
Twin Research and Genetic Epidemiology, King's College London, London, UK. [6]German Center for Cardiovascular Research (DZHK), Partner Site Munich Heart
Alliance, Munich, Germany. [7]Institute for Clinical Diabetology, German Diabetes Center (Deutsches Diabetes-Zentrum/DDZ), Leibniz Center for Diabetes Research at
Heinrich-Heine-University Düsseldorf, Düsseldorf, Germany. [8]Department of Endocrinology and Diabetology, Medical Faculty and University Hospital Düsseldorf,
Heinrich-Heine-University Düsseldorf, Düsseldorf, Germany. [9]German Center for Diabetes Research (DZD), Partner Düsseldorf, Neuherberg, Germany. [10]Deutsches
Herzzentrum München, Technische Universitat München, Munich, Germany. [11]Institute of Epidemiology and Medical Biometry, University of Ulm, Ulm, Germany.
[12]Institute for Medical Informatics, Biometrics and Epidemiology, Ludwig-Maximilians-Universiteit Munchen, Munich, Germany. [13]USDA ARS, Nutrition and Genomics
Laboratory, JM-USDA Human Nutrition Research Center on Aging at Tufts University, Boston, MA, USA. [14]Complex Trait Genetics, Center for Neurogenomics and
Cognitive Research, Vrije Universiteit Amsterdam, Amsterdam, The Netherlands.
✉e-mail: l.j.sinke@lacdr.leidenuniv.nl; b.t.heijmans@lumc.nl

