## [Transparent Peer Review file · Communications Biology]

Epigenome-wide association study of circulating interleukin-6 connects DNA methylation to immunometabolic and inflammatory health

Corresponding Author: Ms Lucy Sinke

Version 1:

Reviewer comments:

Reviewer #1

(Remarks to the Author)

Sinke and coworkers carried out an EWAS metaanalysis of serum IL-6 in three large, well-characterized human cohorts. A number of CpG show significant associations. Those CpG map to regulatory regions and genes involved in inflammation signalling, metabolism and disease risk. Triangulation and MR indicate putative directions of effects between serum IL-6 and methylation of selected CpG. The study is observational and does not experimentally validate any of the EWAS conclusions. Strengths. The sample size (>4,000) is solid. Statistics and bioinformatics are impeccable (but see below). The level of detail provided makes the study reproducible.

Weaknesses. As I read the title of the manuscript, my expectations were very high, but I was disappointed. The work is thorough, but observational and it is a pity that the title boasts conclusions that were not reached. IL-6 is undoubtedly important, but other circulating cytokines mask some of its effects. Hundreds of experimental studies have addressed IL-6 signalling, yet no discussion of that literature, or any attempt to experimentally validate the EWAS data was made. Additionally, the manuscript seems to be written in haste and with little supervision.

Specific points:

- 1) The title is misleading, as it is based solely on gene function enrichment. The title should be more humble and not mechanistic, reflecting the fact that the study is based on observational bioinformatics that must be supported by experimental evidence in IL-6-challenged cellular or animal models.
- 2) Abstract. Introduce the concept of CpG and its relevance in the context of the work. "IL-6 is supported by triangulation as driving methylation....." is one of the many sentences across the manuscript that should be clearer or were written in laboratory jargon. Please carefully revise the text.
- 3) page 8: I find the structure of this segment of the methods confusing. The whole study was a metaanalysis, yet a section named "main analysis" is present, distinct from the section titled metaanalysis on page 10. Consider renaming those sections as "bulk CpG analysis" or similar, and subsequent levels of analysis, all under the metaanalysis umbrella.
- 4) page 8: why were Beta values used in statistics instead of M?
- 5) page 9: I am not aware that "missingness" is a word.
- 6) page 9: what do the Authors mean with "extended cell types"?
- 7) page 9: please briefly explain: "Neutrophils (Neu) were excluded to avoid collinearity (as proportions for all cells sum to 1)".
- 8) Please be consistent in using an abbreviation to indicate IL-6-associated CpG.
- 9) page 10, "Results from each cohort were inspected and rows were removed if they were estimated from fewer than 50 observations": what results were inspected at this preliminary, supposedly CpG-filtering phase? Rows must be individual CpG and stating that would clarify the sentence.
- 10) page 10: please briefly clarify "After running bacon, inflation and bias were estimated at ~1.00 and within ± 0.004 for all models, respectively" for the non-expert reader.
- 11) page 11: please consider a better wording for: "CpG Interpretation".
- 12) page 11: I find the sentence "those with a sample size under 500, those that identified fewer than 100 CpGs at the threshold of inclusion into the respective database," hard to understand.
- 13) page 13: please consider rewriting the sentence: "SuSiE allows evidence for association at multiple causal variants to be evaluated simultaneously, whilst separating the statistical support for each variant conditional on the causal signal being considered, and results in more accurate colocalization inferences".

- 14) page 15: The sentence "This package alongside the noted GWAS was also used to investigate the effects of DNAm at cg26663590 on C-reactive protein..." should not be placed in the methods as written, since the associations with cg26663590 is a result of the study that cannot be anticipated in the methods.
- 15) page 16: please define "PRS -DNAm".
- 16) page 16: "mQTL effect on DNAm": mQTL is by definition an effect on DNAm: could the Authors improve wording?
- 17) Results: showing the methylation Beta range would be useful to appreciate the extent of DNAm changes.
- 18) page 20: I find the sentence "Circulating IL-6 levels have a distinct DNA methylation signature in blood" too strong. The "signature" changes with IL-6 levels, possibly a different term would be more accurate. Then "distinct" from what reference? How is the detected "signature" different or affected by any of other many circulating proinflammatory molecules? CRP was taken into account, but claiming specificity for IL-6 is still a stretch.
- 19) page 20: "blood-based DNA methylation" or blood cell DNA.....?
- 20) page 20: "removal of six CpGs with evidence for heterogeneity of effects between cohorts": please briefly explain the behaviour of those CpG.
- 21) page 21: the abbreviation "IL-6 CpGs" suggests CpG in the IL6 gene. I suggest a better abbreviation (see above).
- 22) page 22: "15 chromatin states" is not consistent with only 14 shown in Fig. 5.
- 23) Fig. 5: any comment of the depleted states?
- 24) page 23: revise the expression "Multi-omic analyses", as only two omics have been exploited.
- 25) page 27, bottom: what is the p associated with rs7185941?
- 26) page 28, bottom: please refer to each panel of Fig. 7 for clarity.
- 27) Discussion: NF-kB is mentioned here, but not emphasized in the Results. Please see above for further comments about the Discussion.

Reviewer #2

(Remarks to the Author)

I really enjoyed reading the paper "DNA methylation links interleukin-6 to the regulation of cytokine signalling, metabolic reprogramming, and downstream disease" by Bas Heijman's group. It is a well conducted meta-analysis EWAS of IL-6 levels, which nicely complements other previous meta-analysis EWAS performed with other inflammatory markers (e.g. CRP). The authors also demonstrate that measuring IL-6 yields associations with DNAm-levels that are independent of the CRP-ones. Overall, there are some very interesting insights gained from this study, including the identification of a CpG that could be causally implicated in driving IL-6 levels and various diseases, although most of the DNAm changes seem to reflect a response to IL-6. This study is an important addition as indeed, it is of critical importance to dissect the DNAm signatures we are seeing in blood into their underlying components, including inflammatory pathways, and to shed insight on potential causal mechanisms.

Most of my comments are technical and if addressed may further improve the manuscript:

- 1) Lines 146-148: I am not sure I understand the rationale for removing samples with IL-6 levels below detection limit, or even those at the opposite extreme. Removing such samples assumes that the authors believe that the measured value (including 0) is random and hence uninformative. However, from my perspective, a value below detection limit, whilst clearly unrealistic, may reflect a low IL-6 value, in which case throwing away the sample seems like a drastic unjustified procedure. Why not cap zero IL-6 values to the minimum non-zero value instead? Same comment applies to outliers in the DNAm data and in the opposite extreme. DNAm or IL-6 outlier values could be capped. In any case, some rational justification for their procedure should be given, or the authors could rerun their analysis using a capped-based procedure to test robustness of conclusions.
- 2) Log-transformation of IL-6: I am not sure if the removal of IL-6 zero levels in the point above is because the authors want to take the log-transformation. If you were to cap IL-6 values to the minimum non-zero value, this avoid the singularity of log-0. However, I would also like to understand why taking log(IL6) is better than just taking IL-6 levels. Could the authors provide some justification for this? Should the authors not do the analysis both ways and test for robustness to conclusions?
- 3) Resolution of cell-type adjustment: I understand that the authors did the analysis two ways: adjusting for 7 immune cell-types and then again for 12 or so. It seems that results were not strongly dependent on this, but I wonder if the authors could provide some rationale as to why they present the primary analysis not adjusting at the higher resolution? Which immune-cell fractions correlate with IL-6?
- 4) I2 threshold seems very high: could the authors also provide some rationale as to why taking $I^2 < 0.9$ is justified? My feeling is that even an I^2 value of 0.5 or 0.6 say would be statistically significant, indicating substantial heterogeneity.
- 5) Substantial age-difference between cohorts: the NTR cohort has a mean age of 38, whilst KORA's mean age is 69 years. This is huge..... Hence, I worry that the great majority of CpGs will have statistically significant I^2 values, because the association of DNAm with IL-6 could well be age-dependent. In any case, this also seems like a missed opportunity. Can the authors explore on potential differences in the associations displayed by the young and old cohorts?
- 6) Chromatin state enrichment: is it worth rerunning this with eFORGE2?

Reviewer #3

(Remarks to the Author)

The manuscript investigates the association between circulating interleukin-6 (IL-6) levels and genome-wide DNA methylation (DNAm) in blood. By performing an epigenome-wide association study (EWAS) on a combined cohort of 4,361 individuals, the study identifies 401 CpG sites associated with IL-6 levels.

The manuscript thoroughly explores the relationship between IL-6 levels and DNA methylation in the studied datasets, identifying CpGs associated with IL-6 levels after correcting for several confounding variables. It provides valuable insights into the interplay between inflammation and DNA methylation. However, while addressing certain statistical and

methodological challenges, there are areas where enhanced clarity and additional data could strengthen the conclusions. Specifically, the suggested causal relationships require further experimental validation. The manuscript would also benefit from a more focused exploration of cell type-specific effects and the incorporation of experimental data. Additionally, the figures do not fully capture the flow of results presented in the text; incorporating more panels could improve the readability and coherence of the visual content.

Specific comments:

1. The Supplementary Tables file (31020_1_supp_0_sqqt6y.docx) does not open in Microsoft Word, so I was unable to review this file.
2. The authors identified 401 CpGs associated with circulating IL-6 levels after multiple filtering steps (line 435), yet no plot is provided to visually assess this association. In addition to the statistical significance already provided, it would be beneficial to include visualizations such as heatmaps (with patients ordered by increasing IL-6 levels), boxplots, or scatter plots to depict DNA methylation versus IL-6 levels.
3. While Figure 1 shows a scatter plot for the sensitivity analysis, a visual representation of the entire pipeline followed, the CpGs obtained, and the plots mentioned in the previous point could greatly enhance readability and understanding.
4. For the EWAS enrichment (lines 218 and 439), can the authors clarify the tissue or cell types in which the different reference EWAS were conducted? EWAS in mixed cell populations like blood can be very noisy, and differences between blood (with a high proportion of neutrophils) and PBMCs can be significant.
5. IL-6 associated CpGs are also linked with CRP levels in previous EWAS. The authors do not discuss the potential association of these CpGs with other cytokine levels in the studied datasets. If such data is available, it could be useful for a better understanding of the specificity of the CpGs to IL-6, especially given the redundancy in immune downstream pathways. Even if no relation is found in the EWAS, this could be due to the lower availability of associations with cytokines in reference to EWAS, so this potential should not be discarded solely on that basis.
6. In Figure 4, the authors show the enrichment of IL-6 associated CpGs in previous EWAS. Several traits such as "Education" or "Aggression" are highly ranked but not interpreted. Additionally, even though the initial covariate adjustment was made for age and sex to derive the IL-6 associated CpGs, these CpGs are also related to both age and sex and should be discussed.
7. The legend of Figure 5 lacks a mention of the background used to compute the odds ratio, and the same is true for the HOMER analysis.
8. In Figure 5b, several TF motifs are enriched in the IL-6 associated CpGs. However, the absence of STAT motifs is notable given their relevance in IL-6 signaling. Can the authors provide any potential explanation for this?
9. The gene set enrichment analysis section (line 530) and associated figure (Figure 6) only display selected pathways. It would be beneficial to list the top gene sets. Additionally, tools like DoRothEA could be used to explore associations with transcription factors.
10. Understanding whether the hypo- or hyper-methylation of IL-6 associated CpGs is a cause or consequence of IL-6 levels is complex. This complexity arises from the potential overlap with other cytokine effects, the analysis's execution in whole blood (which contains different cell types and methylation patterns), and the presence of paracrine and autocrine signaling. Authors might consider using public or newly generated data on ex vivo blood cells treated with IL-6 (e.g., monocytes, neutrophils, PBMCs) to check the overlap between CpGs altered in that process with those found in the study. The authors mention this in the discussion (line 707), but locating such data in literature or generating it would significantly benefit the study.
11. Despite statistical approaches suggesting that cg26663590 might drive disease pathogenesis (line 638) and that its modification could alter risk, this claim seems excessive without supporting experimental data. Additionally, this association has been found in blood, and cannot be tied to any specific cell type. Therefore, I recommend refraining from making such definitive statements without experimental support.

Version 2:

Reviewer comments:

Reviewer #1

(Remarks to the Author)

Dear authors, I am satisfied with the rebuttal and the improvements in the manuscript. Yet, the title is still weak and misleading. Could the following title or a close version work: "EWAS of circulating human IL6 reveals putative links between DNA methylation and inflammation"?

Reviewer #2

(Remarks to the Author)

Biologically, the paper remains interesting. From a technical standpoint however, some major concerns remain, i.e. some of the technical points I had raised have not been properly addressed. Given that this MS claims to have identified some potential causal mediators of IL6 levels, I think it is important that the authors check robustness of their analyses, especially given the lack of experimental validation (as noted by the other 2 reviewers). In more detail, these are the issues that remain outstanding:

1- I was not persuaded by the author's argument to remove undetected IL-6 values as opposed to capping them at a minimum value. The authors response is extremely illogical for the following reasons: (i) if the IL6 distributions are so different between cohorts because of differences in assay-kits, then this will affect ALL of the IL6 values, not just the values at the extremes!, (ii) second, the assay effect is fully confounded with other confounders such as differences in age between cohorts, so the difference in number of detected IL6 levels could be related to age and not technical, (iii) the authors perform a meta-analysis anyway, so this heterogeneity will be reflected in the I^2 values, which I note are huge. This means that setting undetected values of IL6 to NA was not a successful strategy to remove presumed technical heterogeneity between cohorts, (iv) it is simply wrong to set a small undetectable value to NA. If it was undetected, most likely it is because IL6 levels were low. As an analogy, in single-cell RNA-Seq, we don't set 0s to NAs! In sum, the author's procedure is suboptimal and illogical. The capping procedure I had proposed in my earlier comments is logical, specially for IL6 measured with ELISA kits.

2- I am also not persuaded by the authors removing 'outliers' in DNAm. Whilst technical outliers should be removed, there is no evidence whatsoever, based on the data presented, that the outliers being removed in the DNAm data are technical. Moreover, the authors work on the beta-value basis throughout, which is already naturally regularized, so from a statistical perspective, there is no good justification. If the authors had been working on the M-value scale, then beta-values very close to 0 and 1 would become massive outliers in the M-value scale, and in that case I would agree that these would be undesirable for the statistical inference, which is why a capping procedure would then be highly recommended: you don't throw the values away, you cap them, unless you have evidence that they are technical.

3- Meta-analysis is substandard: Apologies for missing this in my first review, but on a 2nd reading I noticed that the authors only used a Fixed-effect meta-analysis. Given the huge I^2 values (I consider I^2 values of 0.5 to be pretty big and indeed their P-values would be very significant!), a fixed effect meta-analysis seems wholly inappropriate. A random-effect model is necessary, or at least authors should report findings based on both a random and fixed effect model. Or is it the case, that after fiddling with the random effect model and not obtaining good results, that they reverted back to only presenting results for the fixed effect model?

4- Related to the previous point, the authors argue that it is a strength that they consider cohorts of widely different demographics. I would only agree with this statement depending on the type of analysis strategy you then adopt. If you perform a fixed-effect meta-analysis with this huge disparity in age-distribution between cohorts, then this is certainly not a strength, but a major weakness. It is disappointing that the authors did not try to assess agreement between the NTR and KORA cohorts, for instance by generating a scatterplot of CpG statistics or effect sizes between each pair of 2 cohorts (3 pairs in total) and demonstrating a correlation between them. This could also reveal potential inconsistencies which however could reflect genuine age-effects of potential biological interest. Currently, it is a critical omission. I strongly advise the authors to add these scatterplots.

5- I thank the authors for adjusting for cell-type heterogeneity at the higher resolution of 12 immune cell-types. I note that a recent preprint (see Fig.1d of the following bioRxiv preprint) Optimizing effect sizes and specificity trumps machine learning when building DNA methylation reference panels for cell-type deconvolution | bioRxiv clearly demonstrates that a proportion of the markers in IDOL-ext DNAm reference panel are far from ideal, as they are characterized by very small effect sizes. The same preprint goes on to rigorously demonstrate that there are better methods to build reliable DNAm reference panels, also at the resolution of 12 immune-cell-types. Hence, given that the IDOL-ext could be suboptimal, I advise rerunning with a more reliable 12 cell-type panel as e.g. implemented in EpiDISH.

Reviewer #3

(Remarks to the Author)

I appreciate the updated manuscript and the responses to my questions. I have no additional comments.

Referee 1: Expert in vascular disease and biology.

Sinke and coworkers carried out an EWAS metaanalysis of serum IL-6 in three large, well-characterized human cohorts. A number of CpG show significant associations. Those CpG map to regulatory regions and genes involved in inflammation signalling, metabolism and disease risk. Triangulation and MR indicate putative directions of effects between serum IL-6 and methylation of selected CpG. The study is observational and does not experimentally validate any of the EWAS conclusions.

Strengths. The sample size (>4,000) is solid. Statistics and bioinformatics are impeccable (but see below). The level of detail provided makes the study reproducible..

Thank you. We are glad that you agree that the level of detail provided in the Methods, alongside Data and Code availability ensures our work was reproducible and thorough.

Weaknesses. As I read the title of the manuscript, my expectations were very high, but I was disappointed. The work is thorough, but observational and it is a pity that the title boasts conclusions that were not reached. IL-6 is undoubtedly important, but other circulating cytokines mask some of its effects. Hundreds of experimental studies have addressed IL-6 signalling, yet no discussion of that literature, or any attempt to experimentally validate the EWAS data was made. Additionally, the manuscript seems to be written in haste and with little supervision.

Although this is not one of the specific points below, we wanted to highlight that we have not approached this rebuttal with haste, and have spent time, love, and care carefully outlining the complex, context-dependent, and well-established nature of inflammatory signalling in more detail in the Introduction (lines 64-73), Results (lines 292-322), and Discussion (lines 519-37).

"Inflammatory cascades are complex and highly context-dependent. IL-6 activation of JAK/STAT signalling, for example, can both amplify IL6 expression in a positive feedback loop and also restrain IL-6 activity via SOCS3. Furthermore, IL-6 production is governed by a multi-layered regulatory network, integrating transcription factors (TFs) with post-transcriptional control to ensure rapid yet reversible induction."

"There were several key limitations to this study. Firstly, the immune response is governed by a complex network of inflammatory proteins, interacting through positive and negative feedback loops. Our analysis, however, was restricted to only two mediators: IL-6, the primary focus, and CRP, which was included as a sensitivity analysis."

Specific points:

1) The title is misleading, as it is based solely on gene function enrichment. The title should be more humble and not mechanistic, reflecting the fact that the study is based on observational bioinformatics that must be supported by experimental evidence in IL-6-challenged cellular or animal models.

We are sad that the title was disappointing and agree that we perhaps overstated the causal nature of our findings. We have proposed a more modest title:

"DNA methylation links with interleukin-6 have implications for cytokine signalling, metabolic reprogramming, and downstream disease."

We have also improved the causal nature of our findings by adding a mediation analysis (lines 377-88) and experimental validation of findings using publicly available data:

1. The conclusion from triangulation (i.e., that identified loci lie downstream of IL-6) was investigated using IL-6 stimulated isolated immune cells (lines 361-7)
2. Links between the CpG identified by 2SMR and expression were validated in two independent datasets where DNAm and expression was profiled in isolated B- and T-cells (lines 413-23; Supplementary Figure 4)

2) Abstract. Introduce the concept of CpG and its relevance in the context of the work. "IL-6 is supported by triangulation as driving methylation....." is one of the many sentences across the manuscript that should be clearer or were written in laboratory jargon. Please carefully revise the text.

We have carefully revised the Abstract (lines 36-47) and are hopeful that the relevance of CpG sites is clearer in this new version. We also steered clear of laboratory jargon, and focus more on the aims of the methods used rather than specific names.

*"Interleukin-6 (IL-6) drives metabolic and inflammatory processes central to disease. Current knowledge implicates epigenetic mechanisms in the regulation of these pathways, **including through the methylation of CpG sites**. This blood-based meta-analysis of three cohorts (n=4,361) identified 401 **IL-6-associated CpGs** enriched in regulatory regions and linked to key immunometabolic genes, including AIM2, MTOR, and IL6R. **Three complementary causal inference approaches** supported most sites as responding to IL-6, with SOCS3 (Suppressor of Cytokine Signalling 3) methylation statistically mediating inflammatory bowel disease risk. Notably, one CpG connected to NFATC2IP (Nuclear Factor of Activated T-cells 2 Interacting Protein) plausibly influenced both IL-6 production and multiple immunometabolic conditions, including body mass index and type 2 diabetes. Collectively, our results map the DNA methylation landscape surrounding circulating IL-6 levels and unveil directional effects and distinct functional relationships between epigenetics and inflammation."*

We have also improved descriptions of specific methods when they arise in the Methods and Results, including triangulation, colocalisation, two-sample Mendelian randomisation, and mediation analyses. For example, for the triangulation section of the Results (lines 335-41), we have added the following to the text:

*"This framework assumes that **if** genetically-determined **outcome levels** ("observed effect") **are predominantly driven by an exposure, then they should be predictable from a combination of genetically-determined exposure and exposure-outcome association** ("predicted effect"). The **correlation** between observed and predicted effects then **quantifies the combined support for a given direction across CpGs**, even if there is insufficient power at individual sites. By performing this analysis **bidirectionally**, we **comparatively inferred** whether the data more strongly supported **IL-6 as a cause or consequence of DNAm.**"*

3) page 8: I find the structure of this segment of the methods confusing. The whole study was a metaanalysis, yet a section named "main analysis" is present, distinct from the section titled metaanalysis on page 10. Consider renaming those sections as "bulk CpG analysis" or similar, and subsequent levels of analysis, all under the metaanalysis umbrella.

We have now renamed the Methods sections:

1. **Cohort descriptions:** descriptions of the cohorts (line 559)
2. **Cohort-based analyses:** EWAS performed in individual cohorts (line 609)
3. **Combined cohort analyses:** meta-analysis of the summary statistics (line 652)
4. **Functional annotation:** characterisation of CpGs wrt traits, chromatin, TFs, and genes (line 680)
5. **Causal inference:** 2SMR, mediation, and triangulation analysis (line 777)

We hope that this makes the different phases of our approach clearer.

4) page 8: why were Beta values used in statistics instead of M?

We thank the reviewer for raising this important methodological choice. Both β -values and M-values are frequently used in DNAm analyses, and each has distinct advantages and limitations. β -values are straightforward to interpret biologically and facilitate harmonization across datasets. In contrast, M-values improve variance stabilization and normality, which can be advantageous for statistical modelling, but they are less intuitive for interpretation as they do not directly correspond to methylation percentages.

In our study, we chose β -values for three reasons: first, the CpG sites identified in our analyses were not at extreme methylation values, where heteroskedasticity is most problematic (mean β -value 0.304), and thus statistical properties of the β -values were not unduly compromised. Second, harmonization was essential for integrative analyses, as publicly available resources we incorporated (e.g., mQTL data from GoDMC) were based on β -values. Third, β -values are

widely used in EWAS, including large-scale studies of inflammatory traits, which enhances comparability of our results with prior literature.

We have added the following text to the Methods for clarity (line 619-20):

"β-values were used to improve downstream interpretation and harmonization with previous analyses."

5) page 9: I am not aware that "missingness " is a word.

Perhaps not! It is a word sometimes used in statistics (Mitra et al., 2023) but I cannot find it in the Cambridge Dictionary. Therefore, I reworded this sentence (line 628-9):

*"In some cases, this resulted in a reduction of the sample size as there **were missing values** in the smoking data."*

6) page 9: what do the Authors mean with "extended cell types"?

Cell types that could be predicted using the IDOL extended algorithm (Salas et al., 2022). The sentence has now been reworded for clarity (lines 631-2):

*"To investigate the effect of adjusting for additional cell-types **whose proportions could be estimated from DNAm data** using the IDOL extended algorithm [...]"*

7) page 9: please briefly explain: "Neutrophils (Neu) were excluded to avoid collinearity (as proportions for all cells sum to 1)".

Since the proportions of predicted cell types collectively sum to 1, you are able to calculate the proportion of neutrophils by subtracting all other cell types from 1:

$$Neu = 1 - Bas - Bmem - Bnv - CD4mem - CD4nv - CD8mem - CD8nv - Eos - Mono - NK - Treg$$

Therefore, the model contains information about the proportion of neutrophils from the other cell types, and you would introduce multicollinearity if you were to add in this final cell type making effects difficult to properly estimate (Jonkman et al., 2025).

We have added the following to the text (lines 635-8):

"[...] meaning a model without neutrophils already contains information about neutrophil proportions and their addition would introduce multicollinearity."

8) Please be consistent in using an abbreviation to indicate IL-6-associated CpG.

Thank you for catching this. They are now consistently called IL-6 associated CpGs throughout the manuscript.

9) page 10, "Results from each cohort were inspected and rows were removed if they were estimated from fewer than 50 observations": what results were inspected at this preliminary, supposedly CpG-filtering phase? Rows must be individual CpG and stating that would clarify the sentence.

If a cohort had derived their IL-6 association using fewer than 50 individuals, we deemed this too small of a sample size and removed this result.

We have improved the text to make this clear (lines 653-7):

*"Cohorts provided summary statistics for each CpG, including the mean and standard error (SE) of DNAm levels, size of association with IL-6 along with the SE, nominal p-value, and number of individuals used to derive the IL-6 association. Results from each cohort were inspected and **CpGs were removed if they were estimated from fewer than 50 participants.**"*

10) page 10: please briefly clarify "After running bacon, inflation and bias were estimated at ~ 1.00 and within ± 0.004 for all models, respectively" for the non-expert reader.

Bacon is a Bioconductor package that was used to adjust for bias and inflation in the test statistics (van Iterson et al., 2017). In order to ensure that this process had sufficiently corrected for the identified bias and inflation, we used bacon to estimate these on the adjusted statistics. If estimated bias and inflation were then close to 0 and 1, respectively, we could conclude that minimal bias and inflation remained in the corrected results. Indeed, we found very low estimated bias (absolute value under 0.004) and inflation after adjustment, leading us to feel confident that bias and inflation had been adequately corrected for.

We have added the following to the manuscript for clarity (lines 664-6):

*"After running bacon, inflation and bias **were re-estimated for all models to ensure that observed bias and inflation had been adequately corrected for.**"*

11) page 11: please consider a better wording for: "CpG Interpretation".

We have moved the DMR analysis into the previous section and this section is now called **Functional Annotation** (line 680).

12) page 11: I find the sentence "those with a sample size under 500 , those that identified fewer than 100 CpGs at the threshold of inclusion into the respective database," hard to understand.

This sentence means that we compared our results only to EWAS which we considered to be large-scale and relevant (i.e. those that had a sample size above 500 and reported finding at least 100 CpGs).

We have rephrased this sentence in the paper (lines 683-6):

*"Following loading of data into R, we **filtered results to include only large-scale, peer-reviewed, and relevant findings** (with a PubMed ID, sample size above 500, finding over 100 CpGs at the threshold of inclusion into the respective database, performed in adults, reporting nominal p-values, and using whole blood or leukocyte samples)."*

13) page 13: please consider rewriting the sentence: "SuSiE allows evidence for association at multiple causal variants to be evaluated simultaneously, whilst separating the statistical support for each variant conditional on the causal signal being considered, and results in more accurate colocalization inferences".

Rewritten to be clearer (732-6):

*"SuSiE **improves fine-mapping and colocalisation** by modelling multiple causal variants within a region simultaneously, **rather than assuming a single causal signal**. It separates the statistical evidence supporting each variant by conditioning on other signals, thereby **reducing confounding between correlated variants** and **yielding more reliable colocalisation inferences**."*

14) page 15: The sentence "This package alongside the noted GWAS was also used to investigate the effects of DNAm at cg26663590 on C -reactive protein..." should not be placed in the methods as written, since the associations with cg26663590 is a result of the study that cannot be anticipated in the methods.

Thank you for catching this. We have removed any mention of specific CpGs from the Methods section.

15) page 16: please define "PRS -DNAm".

We have improved description of triangulation both in the Methods (lines 786-97) and Results (lines 334-41):

*"In detail, when looking at the effect of DNAm on IL-6 levels (consequential analysis), the observed effect is the association between the top mQTL and ln(IL-6), extracted from full GWAS summary data. The predicted effect can then be calculated by combining the mQTL effect extracted from GoDMC data (the association effect of +1 effect allele with DNAm, β_{mQTL}) and the EWAS effect (the association effect of +1 ln(IL-6) with DNAm, β_{EWAS}). Since we wanted to predict the effect of +1 effect allele on IL-6, the calculation for the predicted effect was then $\beta_{mQTL}/\beta_{EWAS}$ in this instance. When looking in the reverse direction (i.e., IL-6 as a cause of DNAm), **the observed effect is a PGS, where the effect of two IL-6 associated SNPs on DNAm were weighted by their effect allele frequencies (EAF)**. The predicted effect used equivalent EAF weighting and was calculated by multiplying the effect of the PGS on IL-6 (β_{PGS}) with the EWAS effect (β_{EWAS}). The observed and predicted effects in both directions were then plotted and correlation was assessed using a Pearson correlation coefficient."*

We hope that this has sufficiently improved the clarity of this methodology.

16) page 16: "mQTL effect on DNAm": mQTL is by definition an effect on DNAm: could te Authors improve wording?

We agree that the previous wording was inaccurate.

We have reworded this section, including an improvement to this line (please see above). It now reads (lines 788-90):

*"[...] mQTL **effect** extracted from GoDMC data (**the association effect of +1 effect allele with DNAm, β_{mQTL}**) [...]"*

17) Results: showing the methylation Beta range would be useful to appreciate the extent of DNAm changes.

Thank you for this suggestion. We have added a sentence about these findings and the effect size distribution to the manuscript (lines 132-4):

*"**Effect sizes ranged from -0.0143 to 0.0065**, with most associations indicating an inverse relationship (n=282, 70.3%; Figures 3a-c)."*

We have also added a volcano plot (Figure 3c) alongside the bidirectional Manhattan plot (Figure 3a), and a plot of effect sizes with their 95% confidence intervals (Figure 3b; lines 135-45).

18) page 20: I find the sentence "Circulating IL-6 levels have a distinct DNA methylation signature in blood" too strong. The "signature" changes with IL-6 levels, possibly a different term would be more accurate. Then "distinct" from what reference? How is the detected "signature" different or affected by any of other many circulating proinflammatory molecules? CRP was taken into account, but claiming specificity for IL-6 is still a stretch.

We agree that our previous wording was too strong and have changed the title of this section (line 85):

*"Circulating IL-6 levels **are associated** with DNA methylation in blood."*

19) page 20: "blood-based DNA methylation" or blood cell DNA.....?

We now more clearly describe the DNAm profiling and motivation for studying IL-6 in blood samples (lines 89-91):

*"Analyses were performed on **DNAm profiled from whole blood samples**; representing an accessible, metabolic tissue widely used in clinical diagnostics and comprised of a mixture of immune cell-types that produce and respond to circulating cytokines."*

20) page 20: "removal of six CpGs with evidence for heterogeneity of effects between cohorts": please briefly explain the behaviour of those CpG.

Added text to clarify (lines 109-113):

*"Six CpGs were excluded due to between-cohort heterogeneity ($I^2 > 90\%$), **comprising three positively and three inversely associated with IL-6**. Although **all showed consistent directions of effect** across cohorts, heterogeneity was increased at these sites by **differences in effect size magnitude** or the **inability to estimate effects in one of the cohorts**."*

21) page 21: the abbreviation "IL-6 CpGs" suggests CpG in the IL6 gene. I suggest a better abbreviation (see above).

Done (see above). We now use the term "IL-6 associated CpGs" throughout the manuscript.

22) page 22: "15 chromatin states" is not consistent with only 14 shown in Fig. 5.

Thank you for catching this. The legend of Figure 5 (now Figure 6 in the revised version) has now been updated (lines 221-5):

*"Figure 6: Functional enrichment of IL-6 associated CpGs. (a) Forest plot of ORs and 95% CIs with enrichment across **14 chromatin states**, calculated using the Roadmap PBMC reference epigenome (E062) with all tested CpGs as the background set. States are sorted by OR, with blue denoting enrichment, yellow indicating depletion, and grey for non-significant results. **One non-significant term (ZNF/Rpts) not shown due to an extremely wide 95% CI.**"*

The data for the excluded term is available in Supplementary Table 6. We hope that this is now sufficiently clear.

23) Fig. 5: any comment of the depleted states?

We have now added text mentioning the depleted states (lines 205-18):

*"Conversely, **three repressive chromatin states were depleted**, including polycomb repressed regions ($OR=0.48$, $p_{fdr}=5.8 \times 10^{-3}$) and bivalent transcription start sites ($OR=0.26$, $p_{fdr}=3.5 \times 10^{-2}$). These patterns*

indicated that IL-6 associated CpGs were preferentially located in regulatory and transcribed chromatin, **rather than transcriptionally silenced regions.**"

24) page 23: revise the expression "Multi-omic analyses", as only two omics have been exploited.

We agree that "Multi-omic analyses" was an incorrect term. We now refer to "Integrative analyses" throughout the text, instead.

25) page 27, bottom: what is the p associated with rs7185941?

The two SNPs instrumenting cg26663590 were rs7185941 and rs7186573 with the following characteristics:

SNP	rs7185941	rs7186573
Location (chr:pos)	chr16:29267117	chr16:28587389
Effect/other allele (EAF)	T/G (0.6576)	C/T (0.6276)
GoDMC mQTL effect size	-0.1005	+0.1402
GoDMC mQTL p-value	<0.0001	<0.0001
GoDMC N	30463	47714
IL-6 GWAS effect size	-0.0098	+0.0161
IL-6 GWAS p-value	0.1221	0.0007
IL-6 GWAS N	11492	27717

Details on the p-value of this SNP have been added to the manuscript (lines 401-3):

"This CpG was instrumented by two independent SNPs (rs7185941 and rs7186573), with the causal signal primarily driven by rs7186573 ($p_{rs7186573}=7.5 \times 10^{-4}$ vs. $p_{rs7185941}=1.2 \times 10^{-1}$)."

26) page 28, bottom: please refer to each panel of Fig. 7 for clarity.

Thank you for catching this. We now refer separately to Figure 7a (now Figure 8a in the revised version) on line 355 and Figure 7b (now Figure 8b in the revised version) on line 356.

27) Discussion: NF-kB is mentioned here, but not emphasized in the Results. Please see above for further comments about the Discussion.

We now mention NF- κ B specifically in the results (lines 232-4):

*"Notably, this included **direct regulators of IL6 expression**, such as Atf4 (9 CpGs), Chop (8 CpGs), Nrf2 (3 CpGs), and **subunits of the NF- κ B protein complex** (p50/p52: 5 CpGs; p65: 10 CpGs)."*

In the Results, we also mention finding a gene set enrichment related to NF- κ B (lines 274-8):

*"Notably, although the top five terms included many large and generic sets (Immune System, Disease, and Metabolism), 25 of the enriched pathways were cytokine-specific (9.4%), including **TNF α signalling via NF- κ B** (14 genes, $p_{fdr}=7.2\times 10^{-4}$), IL-2/STAT5 signalling (16 genes, $p_{fdr}=3.9\times 10^{-5}$), and interleukin-6 family signalling (4 genes, $p_{fdr}=2.0\times 10^{-2}$)."*

We have extended the outline of TF related findings in the Discussion to outline their importance more clearly (lines 444-6):

*"These sites were located near **TFBS relevant to IL-6 biology, including the primary transcriptional regulator of IL6, NF- κ B**, as well as mediators of unfolded protein (Atf4, CHOP) and oxidative stress responses (Nrf2)."*

Referee 2: Expert in Analysis of DNA methylation data.

I really enjoyed reading the paper “DNA methylation links interleukin-6 to the regulation of cytokine signalling, metabolic reprogramming, and downstream disease” by Bas Heijman’s group. It is a well conducted meta-analysis EWAS of IL-6 levels, which nicely complements other previous meta-analysis EWAS performed with other inflammatory markers (e.g. CRP). The authors also demonstrate that measuring IL-6 yields associations with DNAm-levels that are independent of the CRP-ones.

Overall, there are some very interesting insights gained from this study, including the identification of a CpG that could be causally implicated in driving IL-6 levels and various diseases, although most of the DNAm changes seem to reflect a response to IL-6. This study is an important addition as indeed, it is of critical importance to dissect the DNAm signatures we are seeing in blood into their underlying components, including inflammatory pathways, and to shed insight on potential causal mechanisms.

Thank you. We are glad that you enjoyed reading our paper and hope that you find our additions to improve it substantially.

Most of my comments are technical and if addressed may further improve the manuscript:

1) Lines 146-148: I am not sure I understand the rationale for removing samples with IL-6 levels below detection limit, or even those at the opposite extreme. Removing such samples assumes that the authors believe that the measured value (including 0) is random and hence uninformative. However, from my perspective, a value below detection limit, whilst clearly unrealistic, may reflect a low IL-6 value, in which case throwing away the sample seems like a drastic unjustified procedure. Why not cap zero IL-6 values to the minimum non-zero value instead? Same comment applies to outliers in the DNAm data and in the opposite extreme. DNAm or IL-6 outlier values could be capped. In any case, some rational justification for their procedure should be given, or the authors could rerun their analysis using a capped-based procedure to test robustness of conclusions.

Our decision not to cap or impute these values was driven by the differences in assay methods and the limits of detection (LoD) across cohorts:

- In LLS, IL-6 was measured in non-fasted plasma using the PeliKine Compact Human IL-6 ELISA kit.
- In KORA, IL-6 was measured in fasted serum using the Human IL-6 Quantikine HS ELISA kit.
- In NTR, IL-6 was measured in fasted EDTA plasma using an UltraSensitive ELISA.

Capping values could therefore have introduced additional heterogeneity and potential bias between cohorts, and we felt that the most appropriate approach was to retain these values as missing or unmeasured.

We have updated the text to outline this decision to the reader (lines 612-5):

*"Any IL-6 measurements below the limit of detection and outlying values for both DNAm and IL-6 (more than three IQR from the nearest quartile) were removed prior to analysis **rather than imputed to avoid introducing bias or heterogeneity due to the differences in assay detection limits or types between cohorts.**"*

2) Log-transformation of IL-6: I am not sure if the removal of IL-6 zero levels in the point above is because the authors want to take the log-transformation. If you were to cap IL-6 values to the minimum non-zero value, this avoid the singularity of log-0. However, I would also like to understand why taking log(IL6) is better than just taking IL-6 levels. Could the authors provide some justification for this? Should the authors not do the analysis both ways and test for robustness to conclusions?

We did not remove zero levels in order to take the log-transformation, but rather because of the assay differences described above. We instead too the log-transformation because the distribution of IL-6 values was highly skewed. Therefore, we transformed it to improve Normality, avoiding common issues that arise when the Normality assumption is violated, such as a disproportionate effect of outlying individuals and unreliable test statistics.

In detail, for LLS, the distribution of IL-6 (left) and transformed IL-6 (right) showed a considerable improvement in Normality:

For NTR, the distribution of IL-6 (left) and transformed IL-6 (right) also showed a considerable improvement in Normality:

In addition, by natural-log transforming IL-6, we ensured our results were harmonised with previous studies. This enabled combination of results with a prior IL-6 GWAS in our triangulation approach, as such a transformation was also used in this study.

We have added our reasoning to the text (lines 618-9):

*"[...] IL-6 values, measured in pg/mL and natural log-transformed **to improve Normality.**"*

3) Resolution of cell-type adjustment: I understand that the authors did the analysis two ways: adjusting for 7 immune cell-types and then again for 12 or so. It seems that results were not strongly dependent on this, but I wonder if the authors could provide some rationale as to why they present the primary analysis not adjusting at the higher resolution? Which immune-cell fractions correlate with IL-6?

While drafting this paper, the IDOL-ext method for predicting an extended panel of 12 immune cell types was published. To ensure our findings reflected this higher-resolution deconvolution, we incorporated it as a sensitivity analysis rather than re-running the main analysis from scratch. After applying this adjustment, a subset of CpGs (n=130, 24.48%) no longer met the significance threshold and were excluded.

Although this approach strengthened confidence in the robustness of the remaining associations, it carried an important caveat: CpGs whose effects were masked by extended cell-type variation in the initial analysis may therefore have not been detected in our study. However, given that adjustment for extended cell types had only a modest impact on overall effect sizes (correlation between pre- and post-adjusted values: R=0.96), we expected this limitation to have had minimal influence on the overall conclusions.

We have updated the Discussion to reflect this limitation (lines 529-34):

"Adjusting for extended cell-type proportions **slightly reduced the number IL-6 associated CpGs** in this study, and there was **strong concordance** in effect estimates before and after this adjustment ($R=0.96$) suggesting that our findings were **largely robust to finer-grained modeling of immune heterogeneity**. Nevertheless, **some initial associations may have been masked** by the lower resolution adjustment in base models and **could be revealed in future studies using higher resolution deconvolution**."

4) I^2 threshold seems very high: could the authors also provide some rationale as to why taking $I^2 < 0.9$ is justified? My feeling is that even an I^2 value of 0.5 or 0.6 say would be statistically significant, indicating substantial heterogeneity.

While an I^2 value of 0.5 or 0.6 would indicate moderate heterogeneity, we chose to use a more lenient threshold ($I^2 < 0.9$). Our aim was to exclude only those CpGs where heterogeneity was high enough to potentially undermine the robustness of our findings. Although lower I^2 values can also reflect meaningful variation, we believe the high I^2 observed in this study was primarily driven by technical and biological differences between cohorts, including variation in assay sensitivity and type, fasting status, and population demographics (see Table 1). Additionally, for some CpGs, an effect could not be estimated in one of the cohorts, which can artificially inflate heterogeneity statistics.

Despite these higher I^2 values, the direction of effect was consistent across nearly all CpGs, with only 4 of 401 (1.0%) showing opposing effects between cohorts. This indicates that, even when effect sizes varied, the underlying relationships remained stable across different experimental conditions and populations. Interestingly, these four discordant CpGs had relatively low I^2 values (0.0, 10.7, 45.9, and 66.5), suggesting that the inconsistency reflected very small or null effects in one cohort rather than a true reversal of association. In contrast, among the nine CpGs where one cohort could not estimate an effect, three showed I^2 values above 80 despite consistent directions of effect in the other two cohorts, illustrating how missing estimates can disproportionately drive high heterogeneity values.

We now explicitly discuss this as a study limitation in the Discussion (lines 542-6):

"Furthermore, the **three cohorts in this study differed in population demographics, fasting status, and assay sensitivity and types**. These factors likely **contributed to higher heterogeneity**. Since the **directions of effect were largely consistent** across cohorts, with fewer than 1% of IL-6 associated CpGs showing discordance ($n=4$), we **adopted a higher heterogeneity threshold, retaining more CpGs** while acknowledging that this may come **at the expense of precision**."

5) Substantial age-difference between cohorts: the NTR cohort has a mean age of 38, whilst KORA's mean age is 69 years. This is huge.... . Hence, I worry that the great majority of CpGs will have statistically significant I^2 values, because the association of DNAm with IL-6 could

well be age-dependent. In any case, this also seems like a missed opportunity. Can the authors explore on potential differences in the associations displayed by the young and old cohorts?

We agree that the cohorts do differ substantially in terms of age range, but hoped that this would be a strength of our study rather than a weakness, whereby we cover a larger range of ages and therefore of IL-6 levels.

We have added a **Supplementary Figure 1**, which shows the cohort-specific estimates for the top 100 CpGs alongside the meta-analysis result (with corresponding 95% CIs). These are coloured by the absolute difference between the cohort and overall estimate. Visually inspecting these plots, we do see some CpGs have apparent age-related effects (e.g. cg0557921). However, for most CpGs this divergence is small and the overall estimate provides a good balance between effects found in the younger cohort (NTR) and that observed in the older ones (LLS and KORA). Additionally, whilst KORA does diverge more substantially the direction of effects are consistent across cohorts for all 100 CpGs.

We hope that our previous addition to the Discussion relating to the heterogeneity statistic sufficiently conveys these differences to the reader (lines 542-6), alongside the new Supplementary Figure:

*“Furthermore, the **three cohorts in this study differed in population demographics, fasting status, and assay sensitivity and types.** These factors likely contributed to higher heterogeneity. Since the directions of effect were largely consistent across cohorts, with fewer than 1% of IL-6 associated CpGs showing discordance (n=4), we adopted a higher heterogeneity threshold, retaining more CpGs while acknowledging that this may come at the expense of precision.”*

6) Chromatin state enrichment: is it worth rerunning this with eFORGE2?

Thank you for this nice suggestion. We reran the analysis using eFORGE2, finding some comparable results (i.e. enrichment for enhancers across tissues). These results are now included in the manuscript (lines 213-8), as Supplementary Figures 2-3, and with full results in Supplementary Table 7.

Referee 3: Expert in Bioinformatics of omics data including RNA-seq, ChiP-seq, proteomics, and epigenomics.

The manuscript investigates the association between circulating interleukin-6 (IL-6) levels and genome-wide DNA methylation (DNAm) in blood. By performing an epigenome-wide association study (EWAS) on a combined cohort of 4,361 individuals, the study identifies 401 CpG sites associated with IL-6 levels.

The manuscript thoroughly explores the relationship between IL-6 levels and DNA methylation in the studied datasets, identifying CpGs associated with IL-6 levels after correcting for several confounding variables. It provides valuable insights into the interplay between inflammation and DNA methylation. However, while addressing certain statistical and methodological challenges, there are areas where enhanced clarity and additional data could strengthen the conclusions. Specifically, the suggested causal relationships require further experimental validation. The manuscript would also benefit from a more focused exploration of cell type-specific effects and the incorporation of experimental data. Additionally, the figures do not fully capture the flow of results presented in the text; incorporating more panels could improve the readability and coherence of the visual content.

Specific comments:

1. The Supplementary Tables file (31020_1_supp_0_sqqt6y.docx) does not open in Microsoft Word, so I was unable to review this file.

We are sorry to hear that you were unable to view the Supplementary Materials. We have reuploaded them and hope that you are able to access them this time. Please note that they are Microsoft Excel files.

2. The authors identified 401 CpGs associated with circulating IL-6 levels after multiple filtering steps (line 435), yet no plot is provided to visually assess this association. In addition to the statistical significance already provided, it would be beneficial to include visualizations such as heatmaps (with patients ordered by increasing IL-6 levels), boxplots, or scatter plots to depict DNA methylation versus IL-6 levels.

Thank you for this suggestion. However, substantial differences between the three cohorts, including in age distribution, fasting status, and assay platform, would cause a heatmap of such individual level data to be dominated by cohort effects rather than true biological variation. Moreover, none of the individual cohorts has sufficient statistical power alone to robustly detect IL-6-associated CpGs. For these reasons, we focused on combined estimates resulting from a meta-analysis of cohort-specific results. This enabled us to pool statistical power while accounting for within-cohort variability, thereby reducing bias from between-

cohort heterogeneity and providing a rigorous estimate of signals that are both consistent and replicable across studies.

To provide readers with insight into the underlying cohort-level patterns, however, we now include cohort-specific effect estimates and 95% CIs for the top 100 CpGs in **Supplementary Figure 1**. These results reveal that the older cohort with higher average IL-6 levels (KORA) shows markedly larger effect sizes than the others, demonstrating that individuals with elevated IL-6 levels contribute larger effects to the meta-analysis signal. Importantly, however, all three cohorts exhibit consistent effect directions for these CpGs, indicating that while effect magnitudes are greater in older individuals with higher IL-6 levels, the underlying biological relationships are shared across cohorts.

In addition to this Supplementary Figure, we have also improved the visualisation of our results in Figure 3, showing more clearly the top hits, range of effect sizes, and variability, as well as the mean DNAm for the significant CpGs (lines 135-45).

We hope that you agree that these Figures have considerably improved the visualisation of our main findings.

3. While Figure 1 shows a scatter plot for the sensitivity analysis, a visual representation of the entire pipeline followed, the CpGs obtained, and the plots mentioned in the previous point could greatly enhance readability and understanding.

Thank you for this nice suggestion. We have added a flowchart of the pipeline as **Figure 1** to aid clarity and outline our approach for the reader (line 100-4):

4. For the EWAS enrichment (lines 218 and 439), can the authors clarify the tissue or cell types in which the different reference EWAS were conducted? EWAS in mixed cell populations like blood can be very noisy, and differences between blood (with a high proportion of neutrophils) and PBMCs can be significant.

Only EWAS performed in the following tissues were included in the EWAS enrichment analysis (after careful exploration of tissues available in the EWAS catalog and EWAS atlas).

In EWAS catalog:

1. "Blood" or "blood"
2. "Peripheral blood"
3. "Whole blood" or "whole blood" or "Whole Blood"
4. "Cord blood, whole blood" or "Whole blood, cord blood"(included as it also mentioned whole blood)
5. "Whole blood, heel prick blood spots"
6. "Whole blood, breast tissue" (included as it also mentioned whole blood)
7. "CD4+ T-cells, whole blood" or "CD4+ T-cells, Whole blood" or "Whole blood, CD4+ T cells"
8. "Whole blood, CD4+ T-cells, CD14+ monocytes" or "Whole blood, CD4+ T cells, CD14+ monocytes"
9. "Leukocytes"

In EWAS atlas:

1. "blood"
2. "whole blood"
3. "peripheral blood"
4. "blood spot" or "bloodspot"
5. "buffy coat" or "bufy coat"
6. "leukocyte" or "leukocytes" or "white blood cells"

In addition, a filter to include only EWAS in adults was applied, which we intend to filter out any studies on infants or newborns, and thereby improve the comparability of included EWAS to our study.

The following text is now included 683-6):

*"Following loading of data into R, we filtered results to include only large-scale, peer-reviewed, and relevant findings (with a PubMed ID, sample size above 500, finding over 100 CpGs at the threshold of inclusion into the respective database, performed in adults, reporting nominal p-values, and **using whole blood or leukocyte samples**)."*

5. IL-6 associated CpGs are also linked with CRP levels in previous EWAS. The authors do not discuss the potential association of these CpGs with other cytokine levels in the studied datasets. If such data is available, it could be useful for a better understanding of the specificity of the CpGs to IL-6, especially given the redundancy in immune downstream pathways. Even if no relation is found in the EWAS, this could be due to the lower availability of associations with cytokines in reference to EWAS, so this potential should not be discarded solely on that basis.

Unfortunately, data on other cytokines is not available for analysis. Also, apart from CRP there are no other EWAS on cytokines. We have added this limitation to the Discussion (lines 519-22):

*"There were several key limitations to this study. Firstly, **the immune response is governed by a complex network of inflammatory proteins**, interacting through positive and negative feedback loops. **Our analysis, however, was restricted to only two mediators: IL-6, the primary focus, and CRP, which was included as a sensitivity analysis.**"*

6. In Figure 4, the authors show the enrichment of IL-6 associated CpGs in previous EWAS. Several traits such as "Education" or "Aggression" are highly ranked but not interpreted. Additionally, even though the initial covariate adjustment was made for age and sex to derive the IL-6 associated CpGs, these CpGs are also related to both age and sex and should be discussed.

We agree that we did not discuss the EWAS enrichment results as extensively as we could have in the previous version. Therefore, we have improved the text (lines 179-85):

*“Some enrichments, **such as those for education level and aggression**, may have reflected **interactions with environmental, social, and biological factors**. For instance, **education is often correlated with lifestyle factors**, such as diet, which can influence systemic inflammation, while **aggression may arise as a consequence of neuroinflammatory conditions**. Additionally, although models were adjusted for age and sex, we still observed enrichment for CpGs linked to these variables. This suggested that **some CpGs may have represented signatures shared between age, sex, and IL-6**.”*

7. The legend of Figure 5 lacks a mention of the background used to compute the odds ratio, and the same is true for the HOMER analysis.

We have incorporated a new legend for this Figure (now Figure 6 in the revised version; lines 219-27) and improved the description of this analysis in the Methods (lines 706-8):

*“Figure 6: Functional enrichment of IL-6 associated CpGs. (a) Forest plot of ORs and 95% CIs with enrichment across 14 chromatin states, calculated using the Roadmap PBMC reference epigenome (E062) with all tested CpGs as the background set. States are sorted by OR, with blue denoting enrichment, yellow indicating depletion, and grey for non-significant results. One non-significant term (ZNF/Rpts) not shown due to an extremely wide 95% CI. (b) Bar plot of $-\log_{10}p$ -values from TFBS enrichment analysis, performed using HOMER on sequences within 50bp of IL-6 associated CpGs, **tested against a GC-matched random genomic background**.”*

8. In Figure 5b, several TF motifs are enriched in the IL-6 associated CpGs. However, the absence of STAT motifs is notable given their relevance in IL-6 signaling. Can the authors provide any potential explanation for this?

We recognize the absence of STAT motifs in Figure 5b (now Figure 6b in the revised version) as an unexpected finding, particularly given the central role of STAT factors in IL-6 signaling and our triangulation results indicating that IL-6 drives DNAm at the majority of identified CpGs. While STAT transcription factors, especially STAT3, are well-established mediators of IL-6 responses, our motif analysis instead revealed strong enrichment for other factors such as ATF4, NF- κ B, and Nrf2, all of which are key regulators of stress responses and inflammatory pathways.

However, the lack of STAT motif enrichment may reflect the intrinsically low CpG density of STAT binding motifs (Zhang et al., 2020). Despite the IL-6 associated CpGs not residing close to STAT TFBS, when we performed the TF enrichment you suggest below to assess if the genes we identified are targets of a specific TF, we found that they were enriched for being STAT3

targets. This suggested that IL-6 influences DNAm in regions affecting the regulation of STAT3 target genes, even if these CpGs are not located directly within STAT motifs.

Taken together, we hypothesize that IL-6 may modulate the expression of STAT3 target genes through two complementary mechanisms: direct phosphorylation and activation of STAT3 itself, and indirect modulation of cooperating transcription factors, such as NF- κ B, via DNAm changes at their binding sites. This layered regulatory model highlights the complexity of IL-6 signaling and provides a potential explanation for why STAT motifs were not directly enriched in the TFBS analysis, despite the strong biological role of STAT3 in this pathway.

We have added a more extensive discussion of our STAT3 based findings to the Discussion (lines 509-17):

*“One notable finding was that **STAT3, which directly responds to IL-6 signalling, did not have binding sites enriched within 50bp of IL-6 associated CpGs, but that nevertheless genes linked to these CpGs were enriched in the STAT3 regulon.** This could reflect IL-6 modulation of STAT3 targets through multiple mechanisms, both directly through STAT3 phosphorylation, but also indirectly through **DNAm modulation of other TFBS, such as NF- κ B, which cooperate with STAT3 to regulate target genes.** This interpretation would align with the CpG-poor nature of STAT3 motifs. Alternatively, IL-6 could influence **DNAm via STAT3-independent TFs that share downstream targets with STAT3, or via modulation of methylation enzymes, a mechanism supported by prior experimental evidence.**”*

We hope that this extended Discussion alongside the novel TF enrichment analysis has sufficiently improved our consideration of how STAT TFs relate to our results.

9. The gene set enrichment analysis section (line 530) and associated figure (Figure 6) only display selected pathways. It would be beneficial to list the top gene sets. Additionally, tools like DoRothEA could be used to explore associations with transcription factors.

Thank you for these most helpful suggestions. Figure 7a (line 298-308) now shows the top five enriched pathways. We have also performed a TF enrichment analysis using DoRothEA as you suggest and here we find that STAT3 target genes are enriched in our results, despite DNAm not occurring at STAT3 binding sites (lines 280-90). This lends support to our interpretation above that DNAm effects are indeed related to TFs which affect STAT3 targets, but that DNAm does not generally occur at CpG-poor STAT motifs.

We have also significantly improved the alignment of Figure 7c with the text and results from gene set enrichment. The Figure has been simplified and now mentions pathways and genes highlighted in Figure 7a. We also link these findings with the triangulation analysis, showing that both the pathways and genes identified as well as triangulation support DNAm as lying predominantly downstream of IL-6 signalling, whilst acknowledging the complexity of inflammatory cascades (lines 327-9):

“Although we were able to map genes and pathways on an aetiological cascade predominantly downstream of IL-6, immune and inflammatory responses are inherently complex.”

10. Understanding whether the hypo- or hyper-methylation of IL-6 associated CpGs is a cause or consequence of IL-6 levels is complex. This complexity arises from the potential overlap with other cytokine effects, the analysis's execution in whole blood (which contains different cell types and methylation patterns), and the presence of paracrine and autocrine signaling. Authors might consider using public or newly generated data on ex vivo blood cells treated with IL-6 (e.g., monocytes, neutrophils, PBMCs) to check the overlap between CpGs altered in that process with those found in the study. The authors mention this in the discussion (line 707), but locating such data in literature or generating it would significantly benefit the study.

Whilst we were unable to find DNAm data of IL-6 stimulated isolated immune cells, we have now included a section exploring at the transcriptomic response to IL-6 in isolated immune cells (lines 361-7). The 52 genes that responded to IL-6 stimulation in these cells were then used to identify candidate CpGs for a mediation analysis, where we concluded that two CpGs linked to *SOCS3* may mediate IL-6 effects on downstream disease (lines 377-88). We believe that this addition substantially improves the manuscript and thank the reviewer for their suggestion.

The new Figure based on these results is included here:

11. Despite statistical approaches suggesting that cg26663590 might drive disease pathogenesis (line 638) and that its modification could alter risk, this claim seems excessive without supporting experimental data. Additionally, this association has been found in blood, and cannot be tied to any specific cell type. Therefore, I recommend refraining from making such definitive statements without experimental support.

We agree that the previous text was not sufficiently supported by the evidence. We have updated the text to be less excessive (lines 437-9):

*"In summary, this CpG represented **a site worth investigating** for plausibly driving immunometabolic disease risk through IL-6 dependent mechanisms."*

We have also used DNAm and expression data from isolated CD4+ T-cells and B-cells at four stages of development to validate the CpG-gene links identified for this CpG, which confirmed associations for both connected genes, but only with consistent directions of effects for *NFATC2IP* (lines 413-23).

The following text shows updated Figures and outlines data underlying all Figures.

Figure 1 (new) – Added a flowchart of the study design and key findings following a suggestion from Referee 3.

No data underlies this Figure.

Figure 2 (unchanged, previously Figure 1)

(a) Scatterplot of values from Supplementary Table 2:

- x-axis: column D
- y-axis: column H

(b) Scatterplot of values from Supplementary Table 2:

- x-axis: column D
- y-axis: column L

Figure 3 (two subplots added, previously Figure 2)

(a) Scatterplot of values from Supplementary Table 1:

- x-axis: columns D and E (chromosome and position, respectively)
- y-axis: $\log_{10}(\text{column I})$ signed using column G
- coloured by: column D (blue: odd, yellow: even)

(b) Scatterplot of values from Supplementary Table 1:

- x-axis: column G
- y-axis: column A
- coloured by: column B

(c) Scatterplot of values from Supplementary Table 1:

- x-axis: column G
- y-axis: $-\log_{10}(\text{column I})$

Figure 4 (unchanged, previously Figure 3)

(a) Scatterplot of values from Supplementary Tables 1 and 3:

- **x-axis: Supplementary Table 1, column G**
- **y-axis: Supplementary Table 3, column D**

(b) Scatterplot of values from Supplementary Table 4:

- **x-axis: column D**
- **y-axis: column H**

Figure 5 (data unchanged, aesthetics updated, previously Figure 4)

Scatterplot of values from Supplementary Table 5:

- **x-axis: column C**
- **y-axis: column A**
- **95% confidence intervals (CIs) calculated using column D**

Figure 6 (data unchanged, aesthetics updated, previously Figure 5)

(a) Scatterplot of values from Supplementary Table 6:

- **x-axis: column D**
- **y-axis: column B**
- **95% CIs calculated using column E**

(b) Barplot of values in Supplementary Table 8:

- **x-axis: $-\log_{10}(\text{column B})$**
- **y-axis: column A**

Figure 7 (updated with two new subplots, previously Figure 6)

(a) Barplot of values in Supplementary Table 11:

- x-axis: $-\log_{10}(\text{column F})$
- y-axis: column B

(b) Barplot of values in Supplementary Table 12:

- x-axis: $-\log_{10}(\text{column G})$
- y-axis: column B

(c) Image created in BioRender with no underlying data.

Figure 8 (updated and subplots switched, data unchanged, previously Figure 7)

(a) Scatterplot of Supplementary Table 13:

- x-axis: column F
- y-axis: column E

(b) Scatterplot of Supplementary Table 14:

- x-axis: column F
- y-axis: column E

Figure 9 (new plot underlying new analysis)

(a) Scatterplot of values in Supplementary Table 15:

- x-axis: column B
- y-axis: column A
- 95% CIs calculated using column C

(b) Scatterplot of values in Supplementary Table 16:

- Plots faceted by column B
- x-axis: column D
- y-axis: column A
- 95% CIs calculated using column E

References

- Jonkman, T. H., Consortium, B., Zwet, E. W. van, & Heijmans, B. T. (2025). Probing epigenetic clocks as a rational markers of biological age using blood cell counts. *MedRxiv*, 2025.05.12.25327213. <https://doi.org/10.1101/2025.05.12.25327213>
- Mitra, R., McGough, S. F., Chakraborti, T., Holmes, C., Copping, R., Hagenbuch, N., Biedermann, S., Noonan, J., Lehmann, B., Shenvi, A., Doan, X. V., Leslie, D., Bianconi, G., Sanchez-Garcia, R., Davies, A., Mackintosh, M., Andrinopoulou, E. R., Basiri, A., Harbron, C., & MacArthur, B. D. (2023). Learning from data with structured missingness. *Nature Machine Intelligence* 2023 5:1, 5(1), 13–23. <https://doi.org/10.1038/s42256-022-00596-z>
- Salas, L. A., Zhang, Z., Koestler, D. C., Butler, R. A., Hansen, H. M., Molinaro, A. M., Wiencke, J. K., Kelsey, K. T., & Christensen, B. C. (2022). Enhanced cell deconvolution of peripheral blood using DNA methylation for high-resolution immune profiling. *Nature Communications* 2022 13:1, 13(1), 1–13. <https://doi.org/10.1038/s41467-021-27864-7>
- van Iterson, M., van Zwet, E. W., Heijmans, B. T., 't Hoen, P. A. C., van Meurs, J., Jansen, R., Franke, L., Boomsma, D. I., Pool, R., van Dongen, J., Hottenga, J. J., van Greevenbroek, M. M. J., Stehouwer, C. D. A., van der Kallen, C. J. H., Schalkwijk, C. G., Wijmenga, C., Zhernakova, S., Tigchelaar, E. F., Eline Slagboom, P., ... 't Hoen, P. B. (2017). Controlling bias and inflation in epigenome- and transcriptome-wide association studies using the empirical null distribution. *Genome Biology*, 18(1), 1–13. <https://doi.org/10.1186/S13059-016-1131-9/TABLES/4>
- Zhang, S. C., Wang, M. Y., Feng, J. R., Chang, Y., Ji, S. R., & Wu, Y. (2020). Reversible promoter methylation determines fluctuating expression of acute phase proteins. *ELife*, 9. <https://doi.org/10.7554/ELIFE.51317>